



# Ocean acidification indirectly alters trophic interaction of heterotrophic bacteria at low nutrient conditions

**Thomas Hornick[1], Lennart T. Bach[2], Katharine J. Crawfurd[3], Kristian Spilling[4,5], Eric P. Achterberg[2,6], Corina P. D. Brussaard[3,7], Ulf Riebesell[2], Hans-Peter Grossart[1,8*]**

[1]{Leibniz Institute of Freshwater Ecology and Inland Fisheries (IGB), Experimental Limnology, 16775 Stechlin, Germany}

[2]{GEOMAR Helmholtz Centre for Ocean Research Kiel, Düsternbrooker Weg 20, 24105 Kiel, Germany}

[3]{Department of Biological Oceanography, NIOZ – Royal Netherlands Institute for Sea Research, P. O. Box 59, 1790 AB Den Burg, Texel, The Netherlands}

[4]{Marine Research Centre, Finnish Environment Institute, P.O. Box 140, 00251 Helsinki, Finland}

[5]{Tvärminne Zoological Station, University of Helsinki, J. A. Palménin tie 260, 10900 Hanko, Finland}

[6]{National Oceanography Centre Southampton, European Way, University of Southampton, Southampton, SO14 3ZH, UK}

[7]{Aquatic Microbiology, Institute for Biodiversity and Ecosystem Dynamics, University of Amsterdam, P.O. Box 94248, 1090 GE Amsterdam, The Netherlands}

[8]{Potsdam University, Institute for Biochemistry and Biology, 14469 Potsdam, Maulbeerallee 2, Germany}

Correspondence to: T. Hornick (hornick@igb-berlin.de)



## Abstract

Annually, the oceans absorb about one fourth of the anthropogenically produced atmospheric carbon dioxide ($CO_2$) resulting in a drop in surface water pH, a process termed ocean acidification (OA). Surprisingly little is known about how OA affects physiology as well as food web interactions of heterotrophic bacteria when essential nutrients are limited, since most previous experiments were carried out during productive phases or even after nutrient additions to stimulate algal blooms. Therefore, we conducted an *in situ* large-volume mesocosm (~55 m$^3$) experiment in the Baltic Sea by simulating different fugacities of $CO_2$ (*f*$CO_2$) extending from present to future conditions. The study was carried out after the spring-bloom in July-August to maintain low-nutrient conditions throughout the experiment, which resulted in a small-sized phytoplankton community dominated by picophytoplankton. Several positive as well as negative effects on free-living (FL) and particle-associated (PA) bacterial protein production (BPP) and biovolume (BV) could be related to *f*$CO_2$-induced differences in phytoplankton composition and subsequent the availability of phytoplankton-derived organic matter. However, dynamics of BV and cell-specific BPP (csBPP) of FL heterotrophic bacteria could not be explained exclusively by the availability of phytoplankton-derived organic carbon. The dynamics were also related to enhanced grazing on DNA rich (HDNA) bacterial cells at higher *f*$CO_2$ as revealed by flow cytometry. Additionally, a decoupling of autotrophic production and heterotrophic consumption during the last third of the experiment resulted in low, but significantly higher accumulation of DOC at enhanced *f*$CO_2$. Interestingly we could not detect any consistent and direct *f*$CO_2$-induced effect on BPP, csBPP nor BV of either FL or PA heterotrophic bacteria. In contrast, our results reveal several indirect *f*$CO_2$-induced effects on BPP and bacterial BV with potential consequences for oceanic carbon cycling, in particular in a low nutrient and high *f*$CO_2$ future ocean.

## Key words

Ocean acidification, $CO_2$ enrichment, Baltic Sea, KOSMOS mesocosm experiment, bacterial production, phytoplankton, DOC accumulation



## 1 Introduction

Since the industrial revolution the oceans have absorbed ca. one half of the anthropogenic carbon dioxide ($CO_2$), thereby shifting carbonate chemistry equilibria and pH (Caldeira and Wickett, 2003; Raven et al., 2005; Sabine et al., 2004). During the last decade, the Baltic Sea, experienced a pronounced decrease in pH (~0.1 pH units between 1993 and 2012, International Council for the Exploration of the Sea, 2014). This corresponds to a 30% increase in the concentration of $H^+$ during this period (IPCC, 2007) with potential consequences for organism physiology (Fabry et al., 2008, Taylor et al., 2012). At the same time, autotrophic organisms can be fertilized by an enhanced $CO_2$ availability increasing the production of particulate (POM) and dissolved organic matter (DOM) (Egge, et al., 2009; Hein and Sand-Jensen, 1997; Losh et al., 2012; Riebesell et al., 2007). However, most $CO_2$ enrichment experiments studying natural plankton assemblages under variable nutrient conditions do not reveal a consistent response of primary production to elevated $CO_2$ (e.g. Engel, et al., 2005; Hopkinson et al., 2010; Riebesell et al., 2007). Nevertheless, not only the amount, but also the stoichiometric composition of algal DOM and POM can be affected by changes in $fCO_2$. For example, Riebesell et al. (2007) or Maat et al. (2014) reported an increased stoichiometric drawdown of carbon (C) to nitrogen (N) at higher levels of $fCO_2$, most likely as a result from C-overconsumption (Toggweiler, 1993). Since heterotrophic bacteria greatly depend on phytoplankton derived organic carbon (e.g. Azam, 1998), they will most likely respond to alterations in quantity and quality of phyotplankton derived DOM and POM (e.g. Allgaier et al., 2008; Grossart et al., 2006a). Availability and competition for nutrients, however, can substantially alter $fCO_2$-induced changes in activity and biomass of phytoplankton and subsequently of heterotrophic bacteria. In nutrient-depleted or nutrient-limited systems, bacteria can become restricted in their utilization of phytoplankton derived organic matter, depending on the relative availability of inorganic nutrients (Hoikkala et al., 2009; Lignell et al., 2008; Thingstad and Lignell, 1997). Consequently, a $fCO_2$ dependent increase in inorganic C-availability for autotrophs may not stimulate heterotrophic activity. This decoupling of heterotrophic from autotrophic processes has been termed as a "counterintuitive carbon-to-nutrient coupling" (Thingstad et al., 2008). Consequently, bioavailable dissolved organic carbon (DOC) and particulate organic carbon (POC) could accumulate in nutrient limited oceanic surface waters with profound consequences for nutrient cycling and the oceanic carbon pump (Cauwet et al., 2002; Mauriac et al., 2011; Søndergaard



et al., 2000; Thingstad et al., 1997). Various studies reported on limitation of bacterial growth
by inorganic nutrients in several parts of the Baltic Sea (e.g. Hoikkala et al., 2009; Kivi et al.,
1993; Kuparinen and Heinänen, 1993; Zweifel et al. 1993). Based on these results, we
intended to evaluate effects of enhanced $fCO_2$ on activity and biomass of free-living (FL) as
well as particle associated (PA) bacteria during a relatively low productive period of the year
with low levels of nutrients.

**2  Methods**
**2.1  Experimental setup, $CO_2$ manipulation and Sampling**
Nine floating, pelagic KOSMOS (Kiel Off-Shore Mesocosms for future Ocean Simulations;
Riebesell et al., 2013) were moored on $12^{th}$ June 2012 (day -10 = t-10; 10 days before $CO_2$
manipulation) at 59°51.5´N, 23°15.5'E in the Baltic Sea at Tvärminne Storfjärden on the
south-west coast of Finland. Afterwards, the open mesocosm bags were rinsed and water fully
exchanged with the surrounding water masses for five days. Mesocosms were covered on the
top and bottom with a 3 mm net to exclude larger organisms. At t-5, sediment traps were
attached to the bottom at 17 m depth. Further, the submerged mesocosm bags were pulled up
1.5 m above the water surface, enclosing and separating ~55 $m^3$ of water from the surrounding
Baltic Sea and covered by a photosynthetic active radiation (PAR) transparent roof to prevent
nutrient addition from birds and freshwater input from rain. Additionally, existing haloclines
were removed in each mesocosm as described in Paul et al. (2015), thereby creating a fully
homogeneous water body.
The experiment was conducted between $17^{th}$ June (t-5) and $4^{th}$ August (t43) 2012. $CO_2$
addition was performed stepwise on day t0 after sampling and the following three days to
minimize environmental stress on organisms until reaching the initial fugacity-levels of $CO_2$
($fCO_2$). $CO_2$ addition was repeated at t15 in the upper mixed 7 m to compensate for
outgassing. Different $fCO_2$ treatments were achieved by equally distributing filtered (50 μm),
$CO_2$-saturated seawater into the treated mesocosms as described by Paul et al. (2015). Water
samples throughout the whole water column (0-17m) were collected from each mesocosm and
the surrounding seawater using depth-integrated water samplers (IWS, HYDRO-BIOS, Kiel).



Samples for activity measurements were directly subsampled from the IWS on the sampling
boat without headspace to maintain in-situ $f$CO$_2$ concentrations during incubation.
Unfortunately, three mesocosms were lost during the experiment due to welding faults and
thus unquantifiable water exchanges with the surrounding waters. Therefore, we only refer to
the six remaining mesocosms during this report, using the average $f$CO$_2$ from t1 to t43 to
characterize the different treatments as described in Paul et al. (2015):  365 µatm and 368
µatm (controls); 497 µatm, 821 µatm, 1007 µatm and 1231 µatm $f$CO$_2$, respectively. Detailed
descriptions on the study site, mesocosm deployment and system, performance of the
mesocosm facility throughout the experiment, CO$_2$ addition, carbonate chemistry, cleaning of
the mesocosm bags as well as sampling frequencies of single parameters can be obtained from
the experimental overview by Paul et al. (2015).

## 2.2   Physical and chemical parameters

Physical measurements (i.e. temperature and salinity) were performed using a CTC60M
memory probe (Sea and Sun Technology, Trappenkamp, Germany). For these parameters, the
depth-integrated mean values are presented. Full descriptions of sampling and analyses of
Chl $a$, particulate matter (particulate carbon (TPC), particulate organic nitrogen (PON), total
particulate phosphorus (TPP), biogenic silica (BSi)), dissolved organic matter (DOM
including dissolved organic carbon (DOC), dissolved orgnaic nitrogen (DON), dissolved
organic phosphorous (DOP) as well as dissolved inorganic nutrients (phosphate (PO$_4$$^{3-}$),
nitrate (NO$_3$$^-$)) can be obtained from Paul et al. (2015) and in case of DOP measurements
from Nausch et al. (2015).

## 2.3   Microbial standing stock

Abundance of photoautotrophic cells (<20 µm) and free-living (FL) heterotrophic prokaryotes
(HP) were determined by flow cytometry (Crawfurd et al. 2016).  In short, phytoplankton
were discriminated based on their chlorophyll red autofluorescence and/or phycoerythrin
orange autofluorescence (Marie et al., 1999). In combination with their side scatter signal and
size fractionation, the phytoplankton community could be divided into 6 clusters (Crawfurd et
al. 2016), varying in size from 1 to 8.8 µm average cell diameter. Three groups of
picoeukaryotic phytoplankton (Pico I-III), 1 picoprokaryotic photoautotroph (*Synechococcus*



spp.) and 2 nanoeukaryotic phytoplankton groups were detected. Biovolume (BV) estimations
were based on cell abundance and average cell diameters by assuming a spherical cell shape.
The BV sum of Synechococcus and Pico I-III is expressed as $BV_{Pico}$. The BV sum of Nano I
and II will be referred as $BV_{Nano}$. Abundances of FL HP were determined from 0.5 %
glutaraldehyde fixed samples after staining with a nucleic acid-specific dye (Crawfurd et al.
2016). Unicellular cyanobacteria (*Synechococcus* spp.) contributed at max 10% of the total
counts and, therefore, we use the term heterotrophic prokaryotes (HP). Two groups were
identified based on their low (LDNA) and high (HDNA) fluorescence.
Particle-associated (PA) HP were enumerated by epifluorescence-microscopy on a Leica
Leitz DMRB fluorescence microscope with UV- and blue light excitation filters (Leica
Microsystems, Wetzlar, Germany). Fresh samples were gently mixed to prevent particle
settling and a subsample of 15 mL was filtered on a 0.1-% Irgalan Black coloured 5.0 µm
polycarbonate-filter (Whatman, Maidstone, UK) (Hobbie et al., 1977). Thereafter, filters were
fixed with glutaraldehyde (Carl Roth, Karlsruhe, Germany, final conc. 2 %) and stained for 15
min with 4´6-diamidino-2-phenylindole (DAPI, final conc. 1 µg mL$^{-1}$) (Porter and Feig, 1980)
directly on the filtration device and rinsed twice with sterile filtered habitat water before air-
drying and embedding in Citifluor AF1 (Citifluor Ltd, London, UK) on a microscopic slide
(Rieck et al. 2015). Due to mainly small, equally distributed particles on the filters throughout
the experiment, 15 random unique squares were counted with a magnification of 1000x. Total
number of PA HP was enumerated by subtracting autofluorescent cells from DAPI-stained
cells.
BV was calculated separately for FL and PA HP. For FL HP, we used an average cell volume
of 0.06 µm$^3$ reported by Hagström et al. (1979). BV of PA HP were calculated from
measurements of 1600 cells from 3 different mesocosms (346 µatm, 868 µatm, 1333 µatm) as
well as different time points throughout the experiment (t0, t20, t39) according to Massana et
al. (1997). The resulting average BV of 0.16 µm$^3$ per cell was further used to calculate BV of
PA HP from cell abundances. The BV-sum of both size fractions is expressed as total BV of
HP ($BV_{HP}$). Thereby, cell-numbers of PA HP were interpolated with R (R Core Team, 2014),
using splines, to calculate daily abundances. Further, we use the term "HP" and
"heterotrophic bacteria" synonymously, since heterotrophic bacteria account for the majority
of heterotrophic prokaryotes in surface waters (Karner et al., 2001; Kirchman et al. 2007).



Changes in Chl $a$ and BV of heterotrophic bacteria are dependent on various factors, which
are not necessarily related to each other. Therefore, we have standardized $BV_{HP}$ to total Chl $a$
known as a measurement for phytoplankton biomass (Falkowski and Kiefer, 1985). Thereby,
we express a ratio ($BV_{HP}$ : Chl$a$), describing the distribution of heterotrophic bacterial BV and
phytoplankton biomass in relation to $f$CO$_2$.

## 2.4 Bacterial production and community respiration

Rates of bacterial protein production (BPP) were determined by incorporation of $^{14}$[C]-leucine
($^{14}$C-Leu, Simon and Azam, 1989) according to Grossart et al. (2006a). Triplicates and a
formalin-killed control were incubated with $^{14}$C-Leu (213 mCi mmol$^{-1}$; Hartmann Analytic
GmbH, Germany) at a final concentration of 165 nM, which ensured saturation of the uptake
systems of both FL and PA bacteria. Incubation was performed in the dark at *in situ*
temperature (between 7.8°C and 15.8°C) for 1.5 h. After fixation with 2% formalin, samples
were filtered onto 5.0 µm (PA bacteria) nitrocellulose filters (Sartorius, Germany) and
extracted with ice-cold 5% trichloroacetic acid (TCA) for 5 min. Thereafter, filters were
rinsed twice with ice-cold 5% TCA, once with ethanol (50% v/v), and dissolved in
ethylacetate for measurement by liquid scintillation counting (Wallac 1414, Perkin Elmer).
Afterwards, the collected filtrate was filtered on 0.2 µm (FL bacteria) nitrocellulose filters
(Sartorius, Germany) and processed in the same way as the 5.0 µm filters. Standard deviation
of triplicate measurements was usually <15%. The amount of incorporated $^{14}$C-Leu was
converted into BPP by using an intracellular isotope dilution factor of 2. A conversion factor
of 0.86 was used to convert the produced protein into carbon (Simon and Azam, 1989). Cell-
specific BPP rates (csBPP) were calculated by dividing BPP-rates by abundances of PA and
FL HP.
Community respiration (CR) rates were calculated from oxygen consumption during an
incubation period of 48 hours at *in situ* temperature in the dark by assuming a respiratory
quotient of 1 (Berggren et al., 2012). Thereby oxygen concentrations were measured in
triplicate in 120 mL O$_2$ bottles without headspace, using a fiber optical dipping probe
(PreSens, Fibox 3), which was calibrated against anoxic and air saturated water. Further
descriptions are given by Spilling et al. (2015).





## 2.5 Statistical analyses

We used the nonparametric Spearman´s rank correlation coefficient to measure statistical dependence between variables. Significance is determined as $p < 0.05$. Statistical analyses and visualisation were performed using R 3.1.2. (R Core Team, 2014) and R-package "ggplot2" (Wickham, 2009).

## 3 Results

Paul et al. (2015) defined general phases of the experiment by physical characteristics of the water column (temperature) as well as the first $f\mathrm{CO_2}$ manipulation at t0 (Phase 0 = t-5 to t0, Phase I = t1 to t16, Phase II = t17 to t30, Phase III = t31 to t43). These phases characterize also changes in Chl $a$ concentration and chemical bulk parameters. However, heterotrophic bacteria differed in their response with a variable time delay. Consequently, we divided the experiment into new phases based on changes in activity and BV of heterotrophic bacteria. To provide clarity with respect to other publications of the same study, we termed the following phases: **P1 = t0 to t8, P2 = t8 to t26 and P3 = t26 to t43**. The time between closing of the mesocosms and the first $f\mathrm{CO_2}$-manipulation was defined as Phase P0 = t-5 to t0. P1 describes an initial phase without observed $f\mathrm{CO_2}$-related responses in BPP, csBPP or BV. During P2 several positive as well as negative $f\mathrm{CO_2}$-mediated effects on BPP, csBPP and BV were observed, which could be related to the availability of phytoplankton derived organic carbon and effects of bacterial mortality. The end of P2 is defined by reaching the BV maximum of FL heterotrophic bacteria at t26.

### 3.1 Phytoplankton dynamics

Concentration of Chl $a$ increased after closing of the mesocosms until t5, followed by a decline until the end of P1 (t8) (Figure 1). During P0 and P1 no significant $f\mathrm{CO_2}$ related differences in total concentration of Chl $a$ could be observed. During P2, concentrations of Chl $a$ increased again, driven by increasing BV of nanophotoautotrophs ($BV_{Nano}$) until reaching the respective BV maximum of nanophotoautotrophs as well as Chl $a$ at t16-t17 (Figure 1). Thereby, nanophotoautotrophs yielded significantly lower BV with increasing $f\mathrm{CO_2}$ between t13-17 ($r_s=0.68$, $p<<0.01$, n=30), which was reflected in lower concentrations of Chl $a$ in the 3 highest $f\mathrm{CO_2}$-treated mesososms at the Chl $a$ maximum at t16. Thereafter,



both concentrations of Chl $a$ and $BV_{Nano}$ declined until t22-t28, respectively. During the
whole P2, Chl $a$ was highly positively correlated to $BV_{Nano}$ ($r_s$=0.87, p<<0.01, n=123). From
t22 until the end of the experiment, Chl $a$ yielded overall low, but higher concentrations in the
3 highest $f$CO$_2$-treated mesocosms ($r_s$=0.71, p<<0.01, n=76).
BV of picophotoautotrophs ($BV_{Pico}$) was positively correlated to overall Chl $a$ development
during the initial phases P0 and P1 ($r_s$=0.64, p<<0.1, n=66), but showed a strong negative
correlation to Chl $a$ during P2 and P3 ($r_s$=-0.81, p<<0.1, n=162). Especially after the
breakdown of Chl $a$ at t16/t17, $BV_{Pico}$ increased strongly towards the BV maximum at t24 and
remained constant until the end of the experiment (Figure 1). The increase was mainly driven
by BV of *Synechococcus* spp., which accounted for a generally high proportion of $BV_{Pico}$
(31 ± 2 % to 59 ± 2 %) during this study (Figure S1). All four groups of picophotoautotrophs
distinguished by flow cytometry, however, revealed positive or negative $f$CO$_2$-related effects
on BV (Figure 2). During different periods the smallest sized photoautotroph Pico I (~1 µm)
as well as Pico II showed strong fertilization effects of $f$CO$_2$, whereas *Synechococcus* spp. and
Pico III were not and/or negatively affected by $f$CO$_2$.

## 3.2    Bacterial production (BPP) and biovolume (BV)

Heterotrophic bacterial BV was mainly made up by FL bacteria, as PA bacteria contributed to
only 2 ± 0.7 − 10 ± 0.7 % (mean 4.8 ± 0.6 %) of total bacterial BV. PA bacteria, however,
accounted for a substantial fraction of overall BPP (27 ± 1 − 59 ± 7 %, mean 39 ± 4 %). Both
bacterial size-fractions showed distinct dynamics in BV, BPP and csBPP during the course of
the experiment. Interestingly, we could not reveal any consistent and direct $f$CO$_2$ effect on
BPP, csBPP or BV of FL or PA heterotrophic bacteria. Nonetheless, we observed several
$f$CO$_2$-related differences between the mesocosms in BPP of PA bacteria between t16 and t23
as well as BV, BPP and csBPP of FL bacteria within P2.
During the initial phases P0 and P1 changes in BPP and BV of both bacterial size-fractions
paralleled changes in Chl $a$ and $BV_{Pico}$. Thereby, no significant differences or only weak
correlations in FL and PA bacterial BV as well as BPP and csBPP were observed with
changes in $f$CO$_2$ (Table 1). At t8, however, FL bacterial BPP and csBPP yielded 4-5 times
higher rates in the $f$CO$_2$-treated mesocosms compared to both controls (Figure 3). These
higher FL BPP rates were well reflected in significantly higher BV of FL bacteria with



increasing $f\mathrm{CO_2}$ from t10 to t13 ($r_s$=0.72; p<<0.01; n=24). Between t8-t13, FL bacterial BV
was positively correlated to $\mathrm{BV_{Pico}}$ ($r_s$=0.52, p<<0.01, n=36), but particularly to $\mathrm{BV_{PicoI}}$
($r_s$=0.77, p<<0.01, n=36). Surprisingly, after t13/t14, FL bacterial BV declined only in the
three highest $f\mathrm{CO_2}$-treated mesocosms until t18 (Figure 3). In parallel, BPP of both bacterial
size-fractions increased after the breakdown of Chl $a$ at t16 and yielded significantly lower
rates at higher $f\mathrm{CO_2}$ for PA bacteria ($r_s$=-0.52, p<0.01, n=24) as well as FL bacteria ($r_s$=-0.51,
p=0.01, n=24) between t16 and t26. Standardizing BPP rates to cell abundance, however,
revealed only significantly lower csBPP-rates at higher $f\mathrm{CO_2}$ for FL bacteria during this
period ($r_s$=-0.61, p<0.01, n=24). Although we measured similar responses in BPP for PA and
FL bacteria between t16 and t26, BV of both size-fractions revealed contrasting dynamics
(Figure 3, Figure S2). PA bacterial BV declined with the decay of Chl $a$, whereas FL bacteria
increased strongly in BV, which was positively correlated to BV of picophotoautotrophs until
the end of P2. P3 was characterized by declining BPP rates and BV of heterotrophic bacteria.
FL or PA BPP, csBPP or BV were not or negatively correlated to Chl $a$, BV of
picophotoautotrophs or DOC during this period (Table 1).

## 280    4   Discussion

Although OA and its ecological consequences have received growing recognition during the
last decade (Riebesell and Gattuso, 2015), surprisingly little is known about the ecological
effects on heterotrophic bacterial biomass, production or microbial foodweb interactions at
nutrient depleted or nutrient limited conditions, since most of the experiments were carried
out during the productive phases of the year (e.g. phytoplankton blooms), under eutrophic
conditions (e.g. coastal areas), or even with nutrient additions (Allgaier et al., 2008; Brussaard
et al., 2013; Grossart et al., 2006a; Lindh et al., 2013; Riebesell, 2013). However, large parts
of the oceans are nutrient-limited or experience extended nutrient-limited periods during the
year (Moore et al., 2013). Thus, we conducted our experiment in July-August, when nutrients
and phytoplankton production were relatively low in the northeastern Baltic Sea (Hoikkala et
al., 2009; Lignell et al., 2008). During the study, low nitrogen availability limited overall
autotrophic production (Paul et al., 2015, Nausch et al., 2015). This resulted in a post spring
bloom phytoplankton community, dominated by picophytoplankton, which is known to
account for a large fraction of total phytoplankton biomass in oligotrophic, nutrient poor



systems (e.g. Agawin et al., 2000). Nevertheless, dynamics of Chl *a* revealed two minor
blooms of larger phytoplankton during the first half of the experiment. One developed directly
after the closing of the mesocosms, followed by a second one driven by nanophytoplankton
(Paul et al., 2015). Albeit, picophytoplankton accounted mostly > 50 % of Chl *a* during the
entire experiment (Paul et al., 2015). One reason might be, that picoplanktonic cells are
generally favoured compared to larger cells in terms of resource acquisition and subsequent
usage at low nutrient conditions due to their high volume to surface ratio as well as a small
boundary layer surrounding these cells (Moore et al., 2013; Raven, 1998). However, when
cell size is the major factor determining the access to dissolved nitrogen and phosphorous,
bacteria should be able to compete equally or better with picophytoplankton at low
concentrations (Drakare et al., 2003; Suttle et al., 1990). On the other hand, BV and
production of heterotrophic bacteria are highly dependent on quantity and quality of
phytoplankton-derived organic carbon and usually are tightly related to phytoplankton
development (Attermeyer et al., 2014; Attermeyer et al., 2015; Grossart et al., 2003; Grossart
et al., 2006b; Rösel and Grossart, 2012). Consequently, observed $f$CO$_2$-induced effects on
phytoplankton abundance, phytoplankton losses due to grazing and viral lysis as well as $f$CO$_2$-
related differences in phytoplankton composition altered the availability of phytoplankton-
derived organic matter for FL and PA heterotrophic bacteria (Crawfurd et al., 2016; Paul et
al., 2015). Subsequent, changes in BV and production of both size-fractions in relation to
differences in $f$CO$_2$ were observed. However, we could not reveal any consistent pattern of
$f$CO$_2$-induced effects on the coupling of phytoplankton and bacteria. Changes in BV and
production of heterotrophic bacteria were rather indirectly related to different positive as well
as negative $f$CO$_2$-correlated effects on the phytoplankton during relatively short periods.
These periods, however, comprised phases with high organic matter turnover (e.g. breakdown
of Chl *a* maximum). This notion emphasizes the importance to the oceanic carbon cycle,
especially during long periods of general low productivity. The last phase of the experiment
(P3), however, revealed also a decoupling of autotrophic production and heterotrophic
consumption, leading to relatively low, but still significantly higher accumulation of DOC at
enhanced $f$CO$_2$. Nonetheless, we observed additionally $f$CO$_2$-mediated differences in FL
bacterial BV and cell-specific BPP rates, which could be related to effects of enhanced
bacterial grazing at higher $f$CO$_2$ (Crawfurd et al., 2016). Predicting effects on heterotrophic



bacteria in a future, acidified ocean might consequently depend on several complex trophic
interactions of heterotrophic bacteria within the pelagic food web.

### 4.1    Bacteria-phytoplankton coupling at low nutrient concentrations

Heterotrophic bacteria are important recyclers of autochtonously produced DOM in aquatic
systems and play an important role in nutrient regeneration in natural plankton assemblages
(Kirchman 1994, Brett et al., 1999). When phytoplankton is restricted in growth due to the
lack of mineral nutrients, often a strong commensalistic relationship between phytoplanktonic
DOM production and bacterioplanktonic DOM utilization has been observed (Azam et al.,
1983; Bratbak and Thingstad, 1985). Alterations in either growth conditions of phytoplankton
or DOM availability for heterotrophic bacterioplankton, but also losses of phyto- and
bacterioplankton due to grazing or viral lyses can influence the competition for nutrients and
DOM remineralization (Azam et al., 1983; Bratbak and Thingstad, 1985; Caron et al., 1988;
Sheik et al., 2014). The availability of DOM for heterotrophic bacteria may also change, when
they attach to living algae and organic particles. As a consequence, PA bacteria are often less
affected by nutrient limitation due to the generally higher nutrient availability at particle
surfaces (e.g. Grossart and Simon, 1993). In our study, this was reflected in the relatively high
csBPP rates of PA heterotrophic bacteria throughout the entire experiment. However, PA
heterotrophic bacteria contributed only a minor fraction (maximal $10 \pm 0.7$ %) to the overall
heterotrophic bacterial BV, which is usually reported for oligotrophic or mesotrophic
ecosystems (Lapoussière et al., 2010). Nevertheless, the substantial contribution of PA
heterotrophic bacteria to overall BPP emphasizes their importance, especially during such low
productive periods (e.g. Simon et al., 2002, Grossart, 2010). Generally, PA heterotrophic
bacteria are essential for the remineralization of nutrients from autotrophic biomass, which
would otherwise sink out from surface waters (Cho and Azam, 1988; Turley and Mackie,
1994). Leakage of hydrolysis products as well as attachment and detachment of bacteria to
and from particles stimulate production of the FL bacterial size fraction (Cho and Azam,
1988; Grossart et al., 2003, Smith et al., 1992) as well as equally-sized picophytoplankton,
which would be able to compete with bacteria in terms of nutrient-uptake. During the
breakdown of Chl *a* after t16/t17, both FL heterotrophic bacteria and picophotoautotrophs
benefitted from fresh, remineralized POM and their BV and production greatly increased



(Figure 3, Figure S2). The contrasting dynamics of PA heterotrophic bacteria might be a
result of particle losses via sinking (Turley and Mackie, 1994).

### 4.1    $f$CO$_2$-related effects on bacterial coupling to phytoplankton-derived organic matter

Several previous studies demonstrated that responses of heterotrophic bacteria due to changes
in $f$CO$_2$ were related to phytoplankton rather than being a direct effect of pH or CO$_2$ (e.g.
Allgaier et al., 2008, Grossart et al., 2006). Also during this study, BPP and BV of both
heterotrophic bacterial size-fractions were strongly linked to phytoplankton dynamics and
revealed several indirect responses to $f$CO$_2$, resulting from alterations in phytoplankton
community composition and biomass. One small picoeukaryote (Pico I) with cell-diameters of
~1 µm benefitted from the stepwise CO$_2$ addition, yielding significantly higher growth rates
and BV at higher $f$CO$_2$ after t3 (Crawfurd et al., 2016) (Figure 2). This is in line with a few
recent studies, indicating a positive effect of enhanced $f$CO$_2$ on the abundance of small
picoeukaryotic phytoplankton (Brussaard et al., 2013; Endo et al., 2013; Sala et al., 2015).
After t5, Pico I was controlled by grazing and viral lysis with highest reported viral lysis and
loss rates at t10 and t13, respectively (Crawfurd et al., 2016). Interestingly, viral lysis could
only be observed under high CO$_2$ conditions, but not at ambient CO$_2$ levels, which might be
related to higher Pico I productivity at increased $f$CO$_2$ (Crawfurd et al., 2016). Consequently,
at high $f$CO$_2$ biomass production of FL heterotrophic bacteria was fuelled by bioavailable
organic matter from viral lysis and grazing of algal cells (Brussaard et al., 1995; Brussaard et
al. 2005; Sheik et al., 2014). Thus, fertilization effects in photoautotrophic picoplankton
during CO$_2$-addition and subsequent losses (Crawfurd et al., 2016) resulted indirectly in $f$CO$_2$-
related differences in FL bacterial BV between t8 and t14 due to larger availability of
picophytoplankton-derived DOC.
In parallel a second phytoplankton-bloom developed, mainly driven by nanophytoplankton,
which yielded significantly lower BV at higher $f$CO$_2$ (Crawfurd et al., 2016). This was also
reflected in lower Chl $a$ concentrations at highest $f$CO$_2$ (Paul et al., 2015). During breakdown
of Chl $a$ after t16/t17, both BPP of FL and PA bacteria yielded significantly lower rates at
higher $f$CO$_2$, possibly due to the result of lower amounts of nanophytoplankton-derived
organic carbon. Nonetheless, differences in BV and csBPP dynamics of FL heterotrophic
bacteria between t14 and t26 could not be explained exclusively by the availability of





phytoplankton-derived organic carbon, but were rather caused by higher bacterial losses
mainly due to grazing at enhanced $f$CO$_2$ as reported by Crawfurd et al. (2016).

### 4.2  Consequences of $f$CO$_2$-related differences in bacterial mortality for trophic relationships

Not only heterotrophic bacterial activity but also mortality plays an important role in nutrient
regeneration in natural plankton assemblages (e.g. Caron 1994). Two major factors
determining bacterial mortality are viral lysis and grazing (e.g. Liu et al., 2010). The viral
shunt generates mainly bioavailable DOM and stimulates autotrophic and heterotrophic
microbes simultaneously. Advantages in competition for dissolved organic nutrients will
primarily benefit heterotrophic bacteria (e.g. Joint et al., 2002). In contrast, the consumption
of bacterial biomass by bacterivory may release phytoplankton from competition with bacteria
for limiting nutrients (e.g. Bratbak and Thingstad, 1985, Caron et al., 1990). Additionally,
carbon is directly transferred to higher trophic levels (Atkinson, 1996; Sherr et al., 1986;
Schnetzer and Caron, 2005). Both will certainly impact the tight phytoplankton-bacteria
coupling at low nutrient concentrations. However, possible effects of increased $f$CO$_2$ on the
impact of bacterial grazing for trophic interactions are so far largely unknown. Only a few
studies have reported on bacterial grazing in ocean acidification research under different
nutrient conditions and indicated both no effects as well as effects of $f$CO$_2$ (e.g. Brussaard et
al., 2013; Rose et al., 2009; Suffrian et al., 2008).
During our study FL heterotrophic bacterial BV surprisingly dropped only in the highest
$f$CO$_2$-treated mesocosms after t13/t14 and stayed low until t22. In particular, the delay of FL
bacterial BV increase after the Chl $a$ break-down at t16/t17 was rather long, since
heterotrophic bacteria usually react on much shorter time scales to alterations in
phytoplankton-derived organic matter (e.g. Azam et al., 1993). Crawfurd et al. (2016),
however, reported significantly higher bacterial grazing at enhanced $f$CO$_2$ from grazing assays
at t15. Consequently, higher availability of DOM after the decay of the phytoplankton bloom
did stimulate BPP, but this biomass production was directly channelled to a larger proportion
by grazing to higher trophic levels at enhanced $f$CO$_2$ (Atkinson, 1996; Schnetzer and Caron,
2005; Sherr et al., 1986). Nevertheless, we also may add viral lysis here as a possibility for a
higher bacterial mortality. Indeed, viral abundance was higher at enhanced $f$CO$_2$ but increased
already after t8 and remained on a constant level until t22 (Crawfurd et al., 2016). Although it



is unlikely that viral lysis caused the observed $f\text{CO}_2$-related differences in bacterial BV
dynamics between t13/t14 and t26, it still might have added to some of the $f\text{CO}_2$-related
effects during this period.
In addition, Crawfurd et al. (2016) reported following flow cytomety analysis an
accompanying drop of HDNA, but not LDNA bacteria between t13/t14 and t19, which altered
finally the proportion of HDNA:LDNA bacteria in relation to $f\text{CO}_2$ between t14 and t26.
Differentiation of LDNA and HDNA bacteria according to the cell's nucleic acid content can
indicate differences in cell size (Gasol and del Giorgio, 2000), but is more likely a measure
for the cell's activity (Gasol and del Giorgio, 2000; Lebaron et al., 2001; Schapira et al.,
2009). Although we cannot draw any conclusion, if cell size or cell-activity was finally the
determining factor, preferential grazing on HDNA heterotrophic bacteria seems likely (Gasol
et al., 1999, Hahn and Höfle, 2001; Vaqué, 2001). This resulted, however, in a higher
contribution of LDNA and possibly smaller as well as less active cells to the heterotrophic
bacterial population. At higher $f\text{CO}_2$ subsequent FL cell-specific BPP rates were reduced and
BPP maxima more delayed in time between t16 and t26.
Unfortunately, we are not able to relate our results to any possible group of grazing
organisms. Nevertheless, results from Flow Cytometry and counting of protozoa as well as
mesozooplankton indicated possible grazers (Bermúdez et al., 2016, Crawfurd et al., 2016,
Lischka et al., 2015). Bermúdez et al. (2016) reported highest biomass of protozoans around
t15. Biomass was thereby substantially made up by the heterotrophic choanoflagellate
*Calliacantha natans* (Bermúdez, pers. comm.). *Calliacantha natans* was demonstrated to feed
in a size-selective mode only on particles < 1 µm in diameter (Marchant and Scott, 1993) and
thus could be a possible predator on heterotrophic bacteria. Additionally, Crawfurd et al.
(2016) distinguished one group of phototrophic picoeukaryotes by flow cytometry (Pico II),
which only increased in BV and thereby yielded significantly higher BV at higher $f\text{CO}_2$
during the period, when abundance of HDNA bacteria was reduced due to grazing. Although
we do not have any evidence for grazing of both particular groups of organisms, the type of
nutrition would have implications for trophic interactions. If the dominant grazers consisted of
mixotrophic organisms and would be able to fix carbon, they may have directly benefited
from increased $\text{CO}_2$ availability (Rose et al., 2009). Consequently, grazing on bacteria by
mixotrophs might have acted as a direct conduit for primary productivity supported by the use




of inorganic nutrients, which would otherwise be unavailable and bound in bacterial biomass
(Hartmann et al., 2012; Mitra et al. 2014; Sanders, 1991).

### 4.3 Decoupling of $f$CO$_2$-related effects on autotrophic production from bacterial consumption during P3

Exudation of carbon-rich substances by phytoplankton is one of the major sources of labile
DOM for heterotrophic bacteria (Larsson and Hagström, 1979). Exudation is highest under
nutrient-poor conditions, when nutrient limitation impedes phytoplankton growth, but not
photosynthetic carbon fixation (Fogg, 1983). Reported $f$CO$_2$-related increases in primary-
production or in the consumption of inorganic carbon relative to nitrogen (e.g. Riebesell et al.,
1993, Riebesell et al., 2007) may potentially enhance exudation and subsequently alter
phytoplankton-bacteria interactions at higher $f$CO$_2$ (de Kluijver et al., 2010). During the last
phase of the experiment (P3) we indeed observed relatively low, but still significantly higher
DOC accumulation at enhanced $f$CO$_2$ (Figure 4). Although Spilling et al. (2016) could not
reveal any significant differences in primary production due to $f$CO$_2$, also pools of Chl $a$ and
TPC as well as C:N$_{POM}$ showed positive effects related to $f$CO$_2$ (Paul et al., 2015). However,
BPP and heterotrophic bacterial BV of both size-fractions did not reveal any similar $f$CO$_2$-
related differences to DOC concentration or phytoplankton dynamics. This could lead to the
assumption, that heterotrophic bacteria were restricted in growth during P3. Similar findings
have been previously described by other studies, which reported on DOC-accumulation
caused by a limitation of DOM in surface waters (Cauwet et al., 2002; Larsen et al., 2015;
Mauriac et al., 2011; Thingstad et al., 1997, Thingstad et al., 2008). However, generally
strong increase in viral abundance and higher reported viral lysis of several phytoplankton
groups at higher $f$CO$_2$ would have also generated fresh bioavailable DOM during this period
(Crawfurd et al., 2016). Additionally, larger zooplankton increased strong in BV (Lischka et
al., 2015). Therefore an accumulation of DOC by escaping bacterial utilization seems likely,
since heterotrophic bacteria were possibly controlled by viral lysis and grazing. Nevertheless,
remineralized nutrients and carbon from the breakdown of the earlier phytoplankton blooms
were bound to a higher extend in autotrophic biomass at higher $f$CO$_2$ (Paul et al., 2015). This
is also reflected in a lower ratio of BV$_{HP}$ : Chl$a$ with increasing $f$CO$_2$ (Figure 5). However,
during P3 $f$CO$_2$-related differences did not impact sinking flux (Paul et al., 2015). This was
probably related to the domination of small-sized unicellular phytoplankton, which only



contributed indirectly via secondary processing of sinking material to the carbon export
(Richardson and Jackson, 2007, Paul et al., 2015). On the other hand, total CR rates were
significantly reduced at higher $f$CO$_2$ (Spilling et al., 2015) during P3. Interestingly, this
finding would suggest lower CR at higher DOC concentrations. However, CR was strongly
correlated to heterotrophic bacterial BV and thus reflected in the proportion of BV$_{HP}$ : Chl $a$.
Consequently, the counterintuitive difference in CR during P3 is most likely a result of the
"heterotrophy" of the system, which was lower at higher $f$CO$_2$ (Figure 5).

**5    Conclusion**
Microbial processes can be affected either directly or indirectly via a cascade of effects
through the response of non-microbial groups or changes in water chemistry (Liu et al., 2010).
Our large-volume mesocosm approach allowed us to test for multiple $f$CO$_2$-related effects on
heterotrophic bacterial activity and biovolume dynamics on a near-realistic ecosystem level
by including trophic interactions from microorganisms up to zooplankton. Thereby, we
addressed specifically a nutrient-depleted system, which is representative for large parts of the
oceans in terms of low nutrient concentrations and productivity (Moore et al., 2013). During
most time of the experiment, heterotrophic bacterial productivity was tightly coupled to the
availability of phytoplankton-derived organic matter and thus responded to $f$CO$_2$-related
alterations in pico- and nanophytoplankton biovolume, albeit with contrasting results. So far,
this is the first ecosystem study, which cannot only report on positive, but also on
significantly negative effects of higher $f$CO$_2$ on bacterial production. During the experiment,
bacterial mortality from grazing and viral lysis had a strong impact on bacterial biovolume. In
particular, $f$CO$_2$-induced effects on bacterial grazing and its impact on higher trophic levels
are still poorly understood and have been greatly neglected in ocean acidification research. In
our study, however, there was a period when autotrophic production was decoupled from
heterotrophic consumption, which resulted in a low, but significantly higher accumulation of
DOC, with potential consequences for carbon cycling in the upper ocean. Reasons and
consequences of these findings can unfortunately not be generalized, since we did not perform
specific bioassays to test for limiting nutrients. Thus, we highly encourage implementing such
bioassays during further experiments at low nutrient conditions. Our study reveals a number
of $f$CO$_2$-induced effects, which led to responses in biovolume and productivity of



heterotrophic bacteria. Consequently, complex trophic interactions of heterotrophic bacteria in
the pelagic food web, which can only be successfully addressed in whole ecosystem studies,
seem to be the key for understanding and predicting $f$CO$_2$-induced effects on aquatic food
webs and biogeochemistry in a future, acidified ocean.

### 516  **Acknowledgements**

We thank the KOSMOS team and all of the participants in the mesocosm campaign for
organisation, maintenance and support during the experiment. In particular, we would like to
thank Andrea Ludwig for coordinating the campaign logistics and assistance with CTD
operations and the diving team. Further we thank the Tvärminne Zoological Station for the
opportunity to carry out such a big mesocosm experiment at their research station and
technical support on site. Additionally we acknowledge the captain and crew of R/V *ALKOR*
for their work transporting, deploying (AL394) and recovering (AL397) the mesocosms. The
collaborative mesocosm campaign was funded by BMBF projects BIOACID II (FKZ
03F06550) and SOPRAN Phase II (FKZ 03F0611). CPDB was financially supported by the
Darwin project, the Royal Netherlands Institute for Sea Research (NIOZ), and the EU project
MESOAQUA (grant agreement number 228224).





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



Table 1: Spearman´s rank correlation (Spearman´s rank correlation coefficient $r_s$; p-value; n)
of heterotrophic prokaryotic biovolume ($BV_{HP}$), bacterial protein production (BPP) and cell-
specific BPP of size-fractions I) 0.2-5.0 µm (free-living; FL) and II) >5.0 µm (particle-
associated; PA) with $f$CO$_2$, dissolved organic carbon (DOC), community respiration (CR),
chlorophyll $a$ (Chl $a$) and total as well as group-specific biovolumes of pico- and
nanophotoautotrophs (*Synechococcus* spp, Pico I-III, Nano I-II) during the different phases of
the experiment. (n.s.- not significant)

| | I) FL size fraction | | | II) PA size raction | | |
|---|---|---|---|---|---|---|
| | $BV_{HP}$ | BPP | csBPP | $BV_{HP}$ | BPP | csBPP |
| **$f$CO$_2$** | P0: - <br> **P1: 0.36; 0.01; 48** <br> P2: n.s. <br> P3: n.s. | P0: - <br> **P1: 0.5; 0.01; 24** <br> P2: n.s. <br> P3: n.s. | P0: - <br> **P1: 0.55; <<0.01; 24** <br> P2: n.s. <br> P3: n.s. | P0: - <br> P1: n.s. <br> P2: n.s. <br> P3: n.s. | P0: - <br> P1: n.s. <br> P2: n.s. <br> P3: n.s. | P0: - <br> **P1: 0.41; 0.05; 24** <br> P2: n.s. <br> P3: n.s. |
| **DOC** | P0: - <br> P1: n.s. <br> P2: n.s. <br> P3: n.s. | P0: - <br> P1: n.s. <br> **P2: 0.65; <0.01; 17** <br> **P3: -0.35; 0.02; 44** | P0: - <br> P1: n.s. <br> P2: n.s. <br> **P3: -0.35; 0.03; 38** | P0: - <br> P1: n.s. <br> P2: n.s. <br> P3: n.s. | P0: - <br> P1: n.s. <br> **P2: 0.49, 0.05; 17** <br> P3: n.s. | P0: - <br> P1: n.s. <br> P2: n.s. <br> P3: n.s. |
| **CR** | **P0: -0.71; <0.01; 12** <br> **P1: 0.58; <<0.01; 42** <br> **P2: 0.64; <<0.01; 106** <br> **P3: 0.59; <<0.01; 36** | P0: n.s. <br> P1: n.s. <br> **P2: 0.72; <<0.01; 36** <br> P3: n.s. | P0: n.s. <br> P1: n.s. <br> **P2: 0.51; <0.01; 36** <br> P3: n.s. | **P0: -0.62; 0.03; 12** . <br> **P1: 0.5; 0.03; 18** <br> **P2: 0.5; <0.01; 36** <br> P3: n.s. | P0: n.s. <br> P1: n.s. <br> **P2: 0.71, <<0.01; 36** <br> P3: n.s. | P0: n.s. <br> P1: n.s. <br> P2: n.s. <br> P3: n.s. |
| **Chl $a$** | P0: n.s. <br> **P1: 0.77; <<0.001; 48** <br> **P2: -0.77;<<0.001; 112** <br> P3: n.s. | **P0: -0.59; 0.04; 12** <br> **P1: 0.48; 0.02; 24** <br> **P2: -0.41; <0.01; 41** <br> P3: n.s. | **P0: -0.89; 0.02; 6** <br> P1: n.s. <br> P2: n.s. <br> P3: n.s. | **P0: -0.65; 0.02; 12** <br> **P1: 0.39; 0.05; 24** <br> P2: n.s. <br> P3: n.s. | P0: n.s. <br> **P1: 0.51; 0.01; 24** <br> **P2: -0.49, <0.01; 41** <br> P3: n.s. | P0: n.s. <br> P1: n.s. <br> **P2: -0.41; 0.01; 41** <br> **P3: -0.31; 0.05; 41** |
| **$BV_{Nano}$** | P0: n.s. <br> P1: n.s. <br> **P2: -0.75; <<0.01; 112** <br> **P3: -0.46; <<0.01; 51** | P0: n.s. <br> P1: n.s. <br> **P2: -0.35; 0.02; 42** <br> n.s. | P0: n.s. <br> P1: n.s. <br> P2: n.s. <br> **P3: 0.35; 0.05; 33** | P0: n.s. <br> P1: n.s. <br> P2: n.s. <br> **P3: -0.32; 0.05; 39** | **P0: 0.83; 0.04; 6** <br> P1: n.s. <br> **P2: -0.44; <0.01; 42** <br> P3: n.s. | P0: n.s. <br> P1: n.s. <br> **P2: 0.34; 0.03; 42** <br> P3: n.s. |
| **$BV_{Pico}$** | **P0: 0.74; <0.01; 12** <br> **P1: 0.79; <<0.01; 48** <br> **P2: 0.91; <<0.01; 112** <br> P3: n.s. | P0: n.s. <br> **P1: 0.52; <0.01; 24** <br> **P2: 0.65; <<0.01; 42** <br> P3: n.s. | P0: n.s. <br> P1: n.s. <br> P2: n.s. <br> P3: n.s. | P0: n.s. <br> **P1: 0.71; <<0.01; 24** <br> **P2: 0.31; 0.04; 42** <br> P3: n.s. | **P0: n.s.** <br> **P1: 0.58; <0.01; 24** <br> **P2: 0.73, <<0.01; 42** <br> P3: n.s. | P0: n.s. <br> P1: n.s. <br> **P2: 0.37; 0.01; 42** <br> P3: n.s. |





| | | | | | | |
|---|---|---|---|---|---|---|
| **BV$_{Syn}$** | **P0: 0.87; <<0.01; 12**<br>**P1: 0.86; <<0.01; 48**<br>**P2: 0.89; <<0.01; 112**<br>P3: n.s. | P0: n.s.<br>P1: 0.5; 0.01; 24<br>**P2: 0.56; <<0.01; 42**<br>**P3: -0.44; <0.01; 38** | P0: n.s.<br>P1: n.s.<br>P2: n.s.<br>**P3: -0.47; <0.01; 33** | P0: n.s.<br>**P1: 0.64; <<0.01; 24**<br>P2: n.s.<br>P3: n.s. | **P0: 0.83; 0.04; 6**<br>**P1: 0.55; <0.01; 24**<br>**P2: 0.55; <<0.01; 42**<br>**P3: -0.5; <0.01; 38** | P0: n.s.<br>P1: n.s.<br>**P2: 0.37; 0.01; 42**<br>P3: n.s. |
| **BV$_{PicoI}$** | **P0: 0.9; <<0.01; 12**<br>**P1: 0.82; <<0.01; 48**<br>**P2:0.36;<<0.01;110**<br>**P3: -0.28; 0.05; 51** | P0: n.s.<br>**P1: 0.64; <<0.01; 24**<br>P2: n.s.;<br>P3: n.s. | P0: n.s.<br>**P1: 0.53; <0.01; 24**<br>P2: n.s.<br>**P3: -0.34; 0.05; 33** | P0: n.s.<br>**P1: 0.6; <0.01; 24**<br>P2: n.s.<br>P3: n.s. | P0: n.s.<br>**P1: 0.65; <<0.01; 24**<br>P2: n.s.<br>P3: n.s. | **P0: 0.83; 0.04; 6**<br>P1: n.s.<br>P2: n.s.<br>P3: n.s. |
| **BV$_{PicoII}$** | **P0: -0.76; <0.01; 12**<br>**P1: 0.6; <<0.01; 48**<br>P2: n.s.;<br>**P3: 0.36; 0.01; 51** | P0: n.s.<br>**P1: 0.54; <0.01; 24**<br>P2: n.s.<br>**P3: 0.46; <<0.01; 38** | P0: n.s.<br>**P1: 0.4; 0.05; 24**<br>P2: n.s.<br>P3: n.s. | P0: n.s.<br>**P1: 0.58; <0.01; 24**<br>**P2: 0.54; <<0.01; 42**<br>P3: n.s. | **P0: 1; <<0.01; 6**<br>**P1: 0.63; <0.01; 24**<br>P2: n.s.<br>P3: n.s. | **P0: 0.94; <0.01; 6**<br>P1: n.s.<br>P2: n.s.<br>P3: n.s. |
| **BV$_{PicoIII}$** | P0: n.s.<br>P1: n.s.<br>**P2: 0.6; <<0.01; 112**<br>P3: n.s. | P0: n.s.<br>P1: n.s.<br>P2: n.s.<br>P3: n.s. | P0: n.s.<br>P1: n.s.<br>**P2: 0.3; 0.05; 42**<br>P3: n.s. | P0: n.s.<br>P1: n.s.<br>**P2: 0.42; <0.01; 42**<br>P3: n.s. | P0: n.s.<br>P1: n.s.<br>**P2: 0.7; <<0.01; 42**<br>P3: n.s. | P0: n.s.<br>P1: n.s.<br>P2: n.s.<br>P3: n.s. |
| **BV$_{NanoI}$** | P0: n.s.<br>**P1: 0.45; <<0.01; 48**<br>**P2: -0.53; <<0.01; 112**<br>**P3: -0.35; 0.03; 51** | P0: n.s.<br>P1: n.s.<br>P2: n.s.<br>P3: n.s. | P0: n.s.<br>**P1: 0.4; 0.05; 24**<br>**P2: 0.44; <0.01; 42**<br>**P3: 0.41; 0.02; 33** | P0: n.s.<br>P1: n.s.<br>**P2: 0.43; <0.01; 42**<br>P3: n.s. | **P0: 1; <<0.01; 6**<br>**P1: 0.4; 0.05; 24**<br>P2: n.s.<br>P3: n.s. | **P0: 0.94; <0.01; 6**<br>P1: n.s.<br>**P2: -0.44; <0.01; 42**<br>P3: n.s. |
| **BV$_{NanoII}$** | P0: n.s.<br>P1: n.s.<br>**P2: -0.76; <<0.01; 112**<br>P3: n.s. | P0: n.s.<br>P1: n.s.<br>**P2: -0.37; 0.02; 42**<br>P3: n.s. | P0: n.s.<br>P1: n.s.<br>P2: n.s.<br>P3: n.s. | P0: n.s.<br>P1: n.s.<br>P2: n.s.<br>P3: n.s. | **P0: 0.81; 0.05; 6**<br>P1: n.s.<br>**P2: -0.46; <0.01; 42**<br>P3: n.s. | P0: n.s.<br>P1: n.s.<br>**P2: -0.34; 0.03; 42**<br>P3: n.s. |

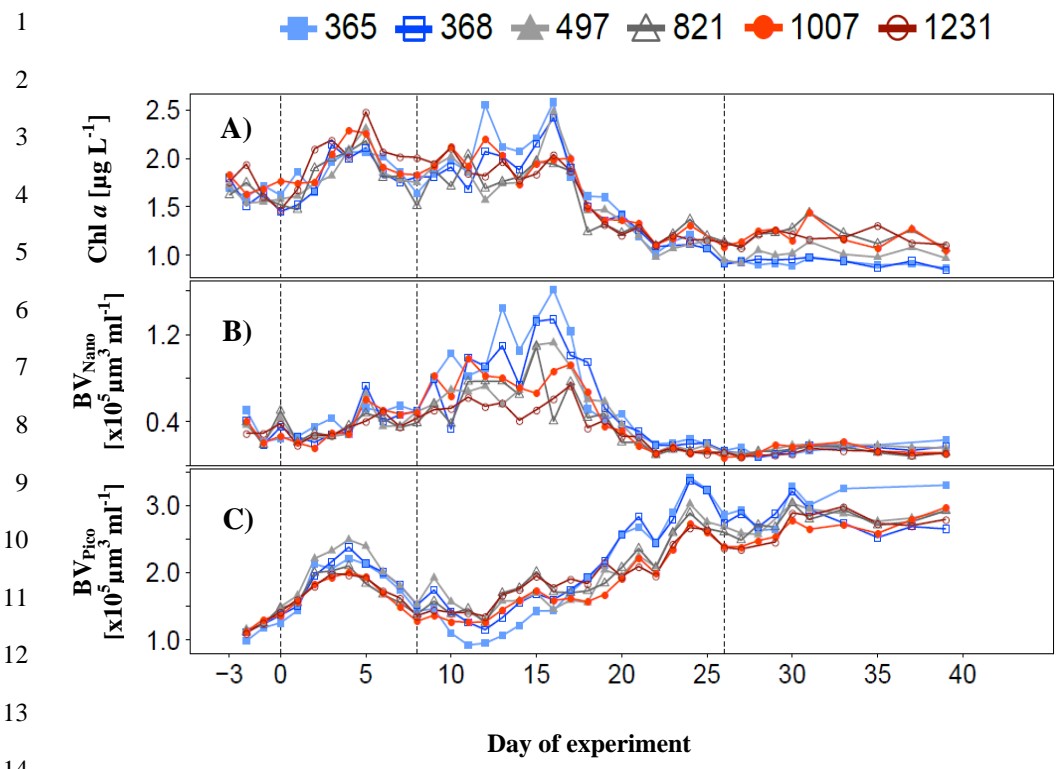

Figure 1. A) Concentration of Chlorophyll $a$ [µg $L^{-1}$], B) biovolume of nanophytoplankton (Nano I and Nano II) [x$10^5$ µm$^3$ ml$^{-1}$] and C) biovolume of picophytoplankton (*Synechococcus* spp., Pico I-III) [x$10^5$ µm$^3$ ml$^{-1}$] during the course of the experiment. Colours and symbols indicate average $f$CO$_2$ [µatm] between t1-t43.





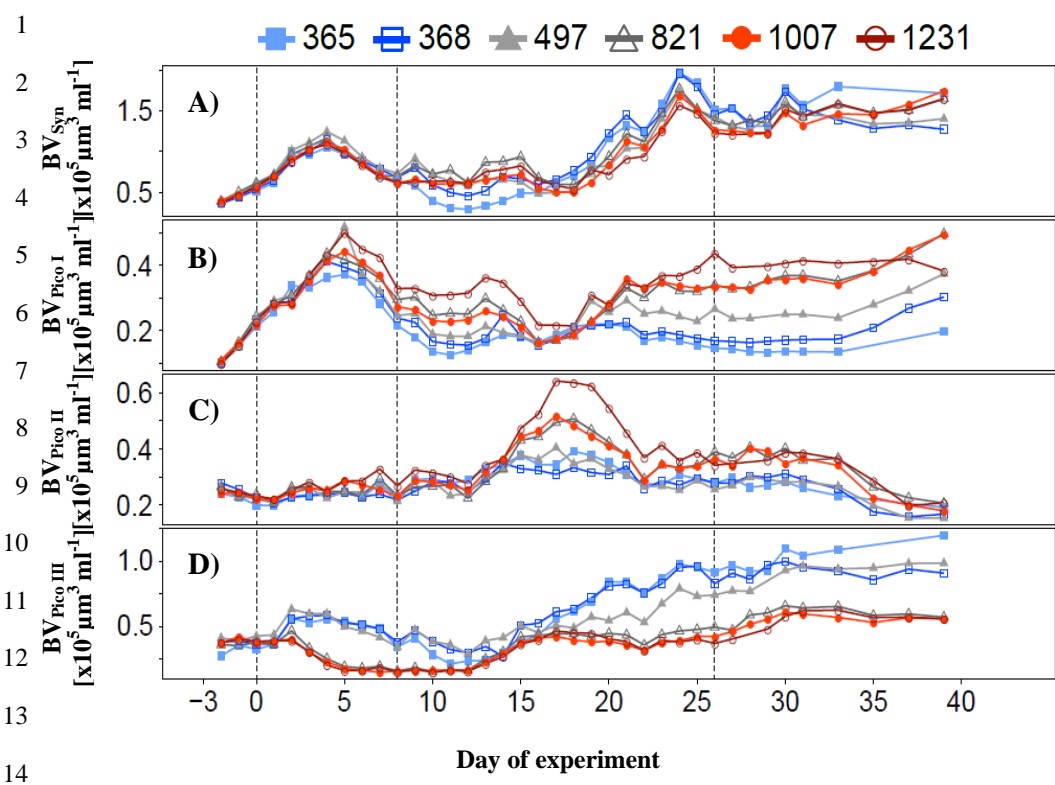

Figure 2. A) biovolume of *Synechococcus* spp. $[x10^5 \, \mu m^3 \, ml^{-1}]$ and B-D) biovolume of picoeukaryote groups I-III (Pico I-III) $[x10^5 \, \mu m^3 \, ml^{-1}]$ during the course of the experiment. Colours and symbols indicate average $f$CO$_2$ [μatm] between t1-t43.








Figure 3. A) Bacterial Protein Production (BPP-C) [$\mu g\,L^{-1}\,d^{-1}$], B) cell-specific Bacterial
Protein Production (csBPP-C) [$fg\,cell^{-1}\,d^{-1}$] and C) biovolume of heterotrophic prokaryotes
($BV_{HP}$) [$x10^5\,\mu m^3\,ml^{-1}$] of size fractions I) 0.2-5.0 µm (free-living bacteria) and II) >5.0 µm
(particle-associated bacteria) during the course of the experiment. D) Ratio of high versus low
nucleic acid stained prokaryotic heterotrophs (HDNA:LDNA), which made up free-living
$BV_{HP}$, revealed from flow cytometry. Colours and symbols indicate average $f$CO$_2$ [µatm]
between t1-t43.



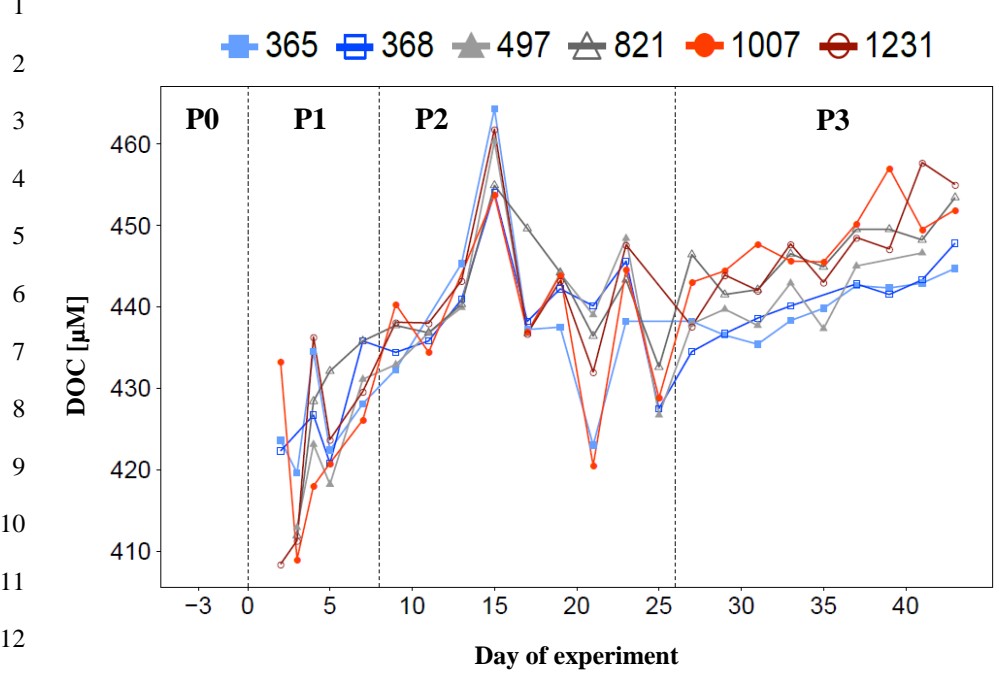

Figure 4. Concentration of dissolved organic carbon (DOC) [µM] during the course of the experiment. During P3, DOC accumulated in the water column, thereby yielding significantly higher concentrations at higher $f$CO$_2$ ($r_s$=0.62; p<<0.01; n=51). Colours and symbols indicate average $f$CO$_2$ [µatm] between t1-t43.



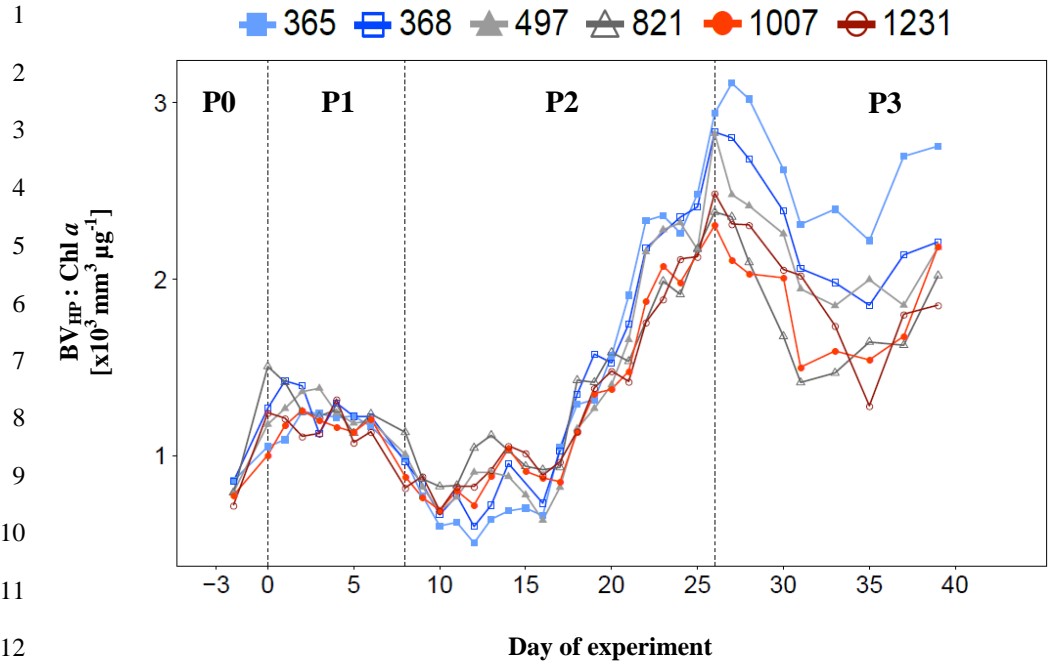

Figure 5. Standardization of heterotrophic prokaryotic biovolume to total Chl $a$ ($BV_{HP}$ : Chl $a$) during the course of the experiment. Colours and symbols indicate average $f$CO$_2$ [μatm] between t1-t43.