# Peer review of "Ocean acidification impacts bacteria-phytoplankton"

_Biogeosciences, 2016_

## Referee Comment (RC1) · Anonymous Referee #1 · 28 Apr 2016

This manuscript addresses an interesting, relevant and timely issue - how bacteria and their C processing may be affected by ocean acidification. As is also pointed out, there are no reasons to expect strong direct effects, while there may be indirect effects channeled through other parts of the food web. This topic is addressed in large scale mesocosms with differing levels of CO2. Unfortunately, I don't find that the manuscript is very clear or efficient in addressing the issue. It is a difficult approach to study a large suite of variables that are to a large extent interdependent and try to understand what has actually happened. In my view, this study shows very minor (if any) effects of CO2 on the bacterial variables measured, and it is hard to clearly link those minor effects to any particular process. Linguistically, I think the manuscript is clear, but I think results

are overstated and relationships over-interpreted, and that the paper lacks a clear focus and structure.

Specific comments:

It is unclear in the title what "trophic interaction" refers to.

There is too little information given to be able to evaluate the methods applied by reading this paper alone. There is a lot of self-referencing to papers covering the same experiment in all parts of the manuscript and this is problematic. Important information that is missing in the methods is for example the dimensions of the mesocosms and the principles behind measuring physical and chemical parameters.

No information is given on the methds behind the estimation of low and high DNA bacteria. Results are included in the figures on low vs high DNA bacteria, but not mentioned in the results text.

It is unclear how statistics were used to show the relationship between e.g. bacterial variables and CO2 within a given time period - how did you account for time within each period?

There is referencing in the results part. Lines 211-218 should be deleted. This manuscript should be able to stand on its own and not make the assumption that we have or will read the other papers from the same experiment. The motivation for dividing into P1 - P3 should be more explicit.

Lines 228-229 "During P2, concentrations of Chl a increased again". I don't think this concurs with the graph.

Lines 236-237 A Spearman rank correlation does not allow to make an interpretation that distinguishes some treatments from others.

Lines 238-240 This negative relationshop between BV of picos and Chl a is puzzling, especially since BV makes out the majority of phytoplankton biomass during the sec-

ond half of the experiment.

Since bacteria are the focus of this manuscript (as I understand the introduction), the results regarding bacteria should be placed first, not phytoplankton. The effects of the treatments on the bacterial variables throughout the experiment are very small. The only statistical effects reported are for P1 and by looking at the graphs (Fig. 3), the relationships with CO2 are hard to discern. Then a few time points are selected and emphasized in the results and discussion because they show differences in relation to CO2 treatments, but they make a out a short period of the experiment.

Figure 4 is not commented on in the results text?

The discussion overall is a little tough to follow, since is not very closely aligned to or focused on the main issue. The discussion shows the difficulties in knowing what a statistical relationship means in this kind of study - the relative role of resource abundance, grazing and viral infections can only be speculated around. Still there are plenty of statements like "...revealed several indirect responses to fCO2, resulting from alterations in phytoplankton community composition and biomass". I am not convinced that the data support such statements.

---

## Referee Comment (RC2) · L. Rhodes (Referee) · 24 Jun 2016

This manuscript is one of a series generated from a mesocosm study conducted in mid- to late summer in 2012 in the Baltic Sea. The time frame selection targeted the period after the spring bloom, when nutrients were expected to be reduced, and the coupling between autotrophic and heterotrophic productivity might be altered. Dissolved CO2 in each mesocosm was manipulated to produce a range of fCO2 from 365 to 1231 $\mu$atm fCO2.

One major concern is the confounding of fCO2 levels and microorganisms added with the CO2-saturated seawater to adjust fCO2 levels. According to Paul et al (2015), different volumes of 50 $\mu$M-filtered seawater were infused in the mesocosms to achieve

a gradient of fCO2. This level of filtration will pass viruses, small grazers, and other microorganisms that can influence trophic interactions. Because the volume of added seawater is correlated with fCO2 levels, it is not possible to separate the abiotic CO2 effect from unknown biotic effects. This confounding problem was not addressed in the manuscript and is a serious problem.

Temperature is a major driver of bacterial abundance and production, but it was not included, even as a covariate, for any analysis. Going back to Paul et al (2015), temperature varied nearly 8°C in a non-monotonic fashion over the experimental period. This important variable should not have been ignored.

Given the number of variables and potential interactions, why wasn't multivariate analysis or similar integrative type of analysis used? Identifying relationships through multiple univariate and bivariate patterns is cumbersome and not necessarily clear to the audience.

Throughout the manuscript, there are references to significant differences in values However, there was only 1 mesocosm per fCO2 level (except for duplicate controls), and no replicate sampling per mesocosm at each time point. There is no information about variation, and therefore, no statistical basis for making statements about significance. Declared differences are based on subjective assessments, rather than objective data analysis.

The discussion could be more succinct and relevant. Much of section 4.2 can be removed, because it is mostly speculative, and ironically, emphasizes the confounding problem mentioned above. This section also contends that grazing was responsible for the drop in bacterial biovolume at higher fCO2, but there is no supporting evidence from this study to support a grazing claim. This is an important point, because the claim is repeated in both the conclusion and abstract.

Related to the decline in bacterial biovolume at higher fCO2 are the actual results, displayed in Figure 2.I.C. Careful examination of that panel in the figure shows that one

of the control mesocosms (368) exhibited a similar decline, for a slightly shorter period of time. In reality, without any information on variation around the data points, it is dangerous to be developing and discussing elaborate explanations of these patterns, if they are even accurate patterns.

Minor points.

Discussion: Numbering for the sections need to be corrected. There is no number for the first portion, and two sections labeled "4.1".

Figure 3. y-axis label for Figure 2.I.B should be for cell-specific BPP.

---

## Author Comment (AC1) · 30 Jul 2016

We kindly thank Linda Rhodes (REVIEWER #2) for her review and taking her time to give constructive comments on our manuscript. We will consider all comments and suggestions when revising the manuscript and have responded below with our comments and description of changes we intend to perform in the revised version of the manuscript. Wherever possible we will incorporate the valuable suggestions. In case, we are not able to follow these suggestions, we hope that we have clearly explained our reasoning.

REVIEWER COMMENT 1: One major concern is the confounding of fCO2 levels and microorganisms added with the CO2-saturated seawater to adjust fCO2 levels. Accord-

ing to Paul et al (2015), different volumes of 50 $\mu$M-filtered seawater were infused in the mesocosms to achieve a gradient of fCO2. This level of filtration will pass viruses, small grazers, and other microorganisms that can influence trophic interactions. Because the volume of added seawater is correlated with fCO2 levels, it is not possible to separate the abiotic CO2 effect from unknown biotic effects. This confounding problem was not addressed in the manuscript and is a serious problem.

Author's response: We are aware of the problem, that a manipulation with CO2-saturated water could impact the planktonic community due to the manipulation itself or the introduced stress by rapid changes in the carbonate system. Therefore, we added CO2-saturated water with the "spider" to rapidly and equally distribute the CO2-saturated water within each mesocosm according to Paul et al. (2015). Moreover, the addition of CO2 was performed in four steps to minimize the stress on the planktonic community by a rapid shift mainly in pH. In addition, reviewer Rhodes pointed out a third issue associated with the addition of CO2-saturated water. As described in Paul et al. (2015), different amounts of 50 $\mu$M prefiltered CO2-saturated water were added to each mesocosm to reveal different fugacities of CO2. However, also the control mesocosms were manipulated with the "spider" and were sparged with prefiltered but not CO2-saturated water (0.04 % of total volume) so that a similar water treatment occurred. Further, the added amounts of CO2-saturated water as compared to the total volume of the mesocosm only contributed to 0.08-0.39 %. A possible seed community, which was introduced by the manipulation with CO2-saturated water made up at maximum 0.35% of the total community. Most of the organisms, however, will die during the preparation of CO2-saturated water. A pH<4 and constant bubbling with CO2 during night will kill most of the organisms, which remained after pre-filtration (own observations). Taking all this into account, the differences in the volume of added CO2-saturated water are to our understanding negligible and will not substantially influence the interpretation of the results. We realize that the text was not clear and thus will be improved in the revised manuscript.
REVIEWER COMMENT 2: Temperature is a major driver of bacterial abundance and production, but it was not included, even as a covariate, for any analysis. Going back to Paul et al (2015), temperature varied nearly 8°C in a non-monotonic fashion over the experimental period. This important variable should not have been ignored.

Author's response: The temperature was similar for all mesocosms and therefore can only potentially have influenced the dynamics of the microbial populations but not the extent of change between the different mesocosms. Nevertheless, the reviewer highlights an important issue, especially when making conclusions on bacterial activity parameters. We included temperature in our revised statistical analysis and will present temperature also in the revised manuscript.

REVIEWER COMMENT 3: Given the number of variables and potential interactions, why wasn't multivariate analysis or similar integrative type of analysis used? Identifying relationships through multiple univariate and bivariate patterns is cumbersome and not necessarily clear to the audience.

Author's response: We agree with reviewer's argument on that and thoroughly revised the statistics using multivariate approaches, i.e. distance-based redundancy analysis (dbRDA) (Legendre and Anderson, 1999) for testing multispecies responses of bacterial activity (bacterial protein production, respiration) and the microbial and phytoplankton community (abundance data) on chemical (dissolved and particulate nutrients) and physical (i.e. temperature) parameters. Thereby, dbRDA results suggest that activity and community was significantly driven by $fCO_2$, pH, temperature and concentrations of total particulate and dissolved organic carbon. In generalized linear and additive modelling we accounted for the temporal correlation (time). Further, we applied network analysis on significant spearman correlation coefficients between Chl a, temperature, $fCO_2$, pH as well as all groups of nano and picophytoplankton, HDNA and LDNA bacteria revealed by flow cytometry (Crawfurd et al., 2015). Thereby, the 3 highest $CO_2$-treated mesocosms clustered significantly different from both controls and the lowest $CO_2$-treated mesocosm (see Figure 1 below).

[Figure]

REVIEWER COMMENT 4: Throughout the manuscript, there are references to significant differences in values. However, there was only 1 mesocosm per fCO2 level (except for duplicate controls), and no replicate sampling per mesocosm at each time point. There is no information about variation, and therefore, no statistical basis for making statements about significance. Declared differences are based on subjective assessments, rather than objective data analysis.

Author's response: The reviewer raised an important point about the statistical analyses of the experiment. However, the experiment was designed to catch a gradient of different levels of CO2 to apply regression analysis or having the opportunity to analyse tipping points of a response to CO2 as well as analysing non-linear responses. We agree that we do not know a within-group variation of a single CO2-treatment but this is not mandatory for regression analyses. Statistically, a regression is equally valid compared, i.e. to an analysis of variance (ANOVA) to making statements about significance. Besides, parameters with possible large measurement-variations or small sample volumes (i.e. bacterial protein production (BPP)) were measured in triplicate to account for the variance within the measurement. For these parameters the mean of 3 measurements is presented. However, since these are pseudo-replicates, there is no additional value for any statistical test. We are aware that a spearman rank correlation is based on the rank and only describes the relationship between two variables by using a monotonic function. Therefore, it is probably not appropriate to make conclusions on multivariate interdependent variables. However, we reanalysed the data and applied more appropriate statistical tests and models like dbRDA (see COMMENT 3).

REVIEWER COMMENT 5: The discussion could be more succinct and relevant. Much of section 4.2 can be removed, because it is mostly speculative, and ironically, emphasizes the confounding problem mentioned above. This section also contends that grazing was responsible for the drop in bacterial biovolume at higher fCO2, but there is no supporting evidence from this study to support a grazing claim. This is an important point, because the claim is repeated in both the conclusion and abstract.

[Figure]

Author's response: As reviewer 2 addresses right, final supporting data for any evidence of a grazing claim is missing. Therefore, we will remove speculative assumptions and incorporate the section 4.2 into other sections of the discussion. The discussion will be reworked accordingly.

REVIEWER COMMENT 6: Related to the decline in bacterial biovolume at higher $fCO_2$ are the actual results, displayed in Figure 2.I.C. Careful examination of that panel in the figure shows that one of the control mesocosms (368) exhibited a similar decline, for a slightly shorter period of time. In reality, without any information on variation around the data points, it is dangerous to be developing and discussing elaborate explanations of these patterns, if they are even accurate patterns.

Author's response: We thank the reviewer for pointing out that this was not examined sufficiently previously in the manuscript. We will rework it.

REVIEWER COMMENT 7: Minor points: Discussion: Numbering for the sections need to be corrected. There is no number for the first portion, and two sections labeled "4.1". Figure 3. y-axis label for Figure 2.I.B should be for cell-specific BPP.

Author's response: These 2 points will be corrected accordingly.

References

Crawfurd, K. J., Brussaard, C. P. D., and Riebesell, U.: Shifts in the microbial community in the Baltic Sea with increasing CO2, Biogeosciences Discuss., doi:10.5194/bg-2015-606, in review, 2016.

Legendre, P. and Anderson, M.J.: Distance-based redundancy analysis: testing multispecies responses in multifactorial ecological experiments. Ecological Monographs, 69, 1-24, 1999.

Paul, A., Bach, L.T., Schulz, K.-G., Boxhammer, T., Czerny, J., Achterberg, E.P., Hellemann, D., Trense, Y., Nausch, M., Sswat, M., and Riebesell, U.: Effect of elevated CO2 on organic matter pools and fluxes in a summer Baltic Sea plankton community,

Biogeosciences, 12, 6181–6203, doi:10.5194/bg-12-6181-2015, 2015.

[Figure]

[Figure]

[Figure]

Figure 1: Heatmap and cluster analyses (euclidean distance) on significant spearman correlation coefficients (r) between Chlorophyll *a*, temperature, all groups of nano and picophytoplankton as well as HDNA and LDNA bacteria revealed by flow cytometry (Crawfurd et al., 2015). Thereby the 3 highest $CO_2$-treated mesocosms clustered significantly different from both controls and the lowest $CO_2$-treated mesocosm (average levels of $f$CO$_2$ t1-t43 are reported for each mesocosm).

**Fig. 1.**

---

## Author Comment (AC2) · 30 Jul 2016

We kindly thank Reviewer #1 for the review and taking the time to provide numerous constructive comments on our manuscript. We will consider almost all comments and suggestions when revising the manuscript and have responded below with our comments and description of changes we will make to the manuscript. Wherever possible we will incorporate the suggestions. In case, we are not able to follow the suggestion, we hope that we have clearly explained our reasoning.

REVIEWER COMMENT 1: This manuscript addresses an interesting, relevant and timely issue - how bacteria and their C processing may be affected by ocean acidification. As is also pointed out, there are no reasons to expect strong direct effects, while

there may be indirect effects channelled through other parts of the food web. This topic is addressed in large scale mesocosms with differing levels of CO2. Unfortunately, I don't find that the manuscript is very clear or efficient in addressing the issue. It is a difficult approach to study a large suite of variables that are to a large extent interdependent and try to understand what has actually happened. In my view, this study shows very minor (if any) effects of CO2 on the bacterial variables measured, and it is hard to clearly link those minor effects to any particular process. Linguistically, I think the manuscript is clear, but I think results are overstated and relationships over-interpreted, and that the paper lacks a clear focus and structure.

Author's response: We acknowledge that reviewer 1 raised these critical points. In contrast to most other studies dealing with effects of ocean acidification, we did not add nutrients to study the effects of changing CO2 on nutrient cycling in a plankton community at naturally low nutrient conditions. The purpose of the experiment was to especially test effects of changes in CO2 on a nutrient limited phytoplankton community and if possible effects on this phytoplankton community can feed back on bacterial activity and abundance. No pronounced direct effects of CO2 on bacterial variables were observed throughout the experiment. Although only minor effects could be observed in this study, the obtained results will be crucial to better understand the role of nutrients on both direct and indirect effects of CO2 on planktonic communities. However, we realize that some reported effects might be overemphasized in our old discussion and thus will reconsider their relevance. In the revised version of the manuscript we will focus better on bacterial aspects and try to link them more specifically to particular processes, supported by a very thoroughly reanalysed statistics (see comments by and our reply to reviewer #2). Consequently, large parts of the manuscript will be revised according to the suggestions of reviewer 1. Further detailed descriptions on changes, which will be amended to the manuscript and will be answered in the following responses on the comments raised by reviewer 1.

REVIEWER COMMENT 2: It is unclear in the title what "trophic interaction" refers to.

Author's response: We realize that the title was not clear in that respect. We will change the title of the revised manuscript and address it as a question to avoid any overstatement of the results mentioned by reviewer 1: "Does ocean acidification alter nutrient- prokaryote-phytoplankton relationships at low nutrient conditions?

REVIEWER COMMENT 3: There is too little information given to be able to evaluate the methods applied by reading this paper alone. There is a lot of self-referencing to papers covering the same experiment in all parts of the manuscript and this is problematic. Important information that is missing in the methods is for example the dimensions of the mesocosms and the principles behind measuring physical and chemical parameters.

Author's response: Thanks for highlighting this important issue. We reduced on purpose as much information as possible, which is given in the core paper by Paul et al. (2015) (i.e. measurements of dissolved and particulate nutrients) to condense our results section and increase the word flow. However, we realize that it might be important to include brief descriptions of measurements of physical and chemical parameters as well as the mesocosm set-up for providing a better background on the experiment, although this was already done in the core paper by Paul et al. (2015). In the revised manuscript we will better specify the methods and try to reduce self-referencing to papers covering the same experiment wherever possible.

REVIEWER COMMENT 4: No information is given on the methods behind the estimation of low and high DNA bacteria. Results are included in the figures on low vs. high DNA bacteria, but not mentioned in the results text.

Author's response: Two groups of heterotrophic prokaryotes were identified based on their low (LDNA) and high (HDNA) fluorescence (Lines 150-151). This identification was based on gating of SYBR green I fluorescence (nucleic-acid specific dye) against the side scatter signal determined by flow cytometry (Brussaard, 2004 with adaptation according to Mojica et al., 2014) as discussed in Crawfurd et al. (2015). We will specify

this in the revised manuscript and will also include data of the ratio between LDNA and HDNA bacteria in the results section.

REVIEWER COMMENT 5: It is unclear how statistics were used to show the relationship between e.g. bacterial variables and CO2 within a given time period - how did you account for time within each period?

Author's response: So far, statistics were solely based on spearman rank correlation. Thereby, we assigned a spearman rank correlation between two variables using all measurements within a given time period. We realize (see rebuttal to reviewer #2), that this might be problematic for interpreting multivariate relationships. We revised the statistics using multivariate approaches, i.e. distance-based redundancy analysis (dbRDA) (Legendre and Anderson, 1999) for testing multispecies responses of bacterial activity (bacterial protein production, respiration) and the microbial and phytoplankton community (abundance data) on chemical (dissolved and particulate nutrients) and physical (i.e. temperature) parameters. Our dbRDA results suggest that activity and community was significantly driven by fCO2, pH, temperature and concentrations of total particulate and dissolved organic carbon. In addition, we applied generalized linear and additive modelling to account for temporal correlations (time).

REVIEWER COMMENT 6: There is referencing in the results part. Lines 211-218 should be deleted. This manuscript should be able to stand on its own and not make the assumption that we have or will read the other papers from the same experiment. The motivation for dividing into P1 - P3 should be more explicit.

Author's response: The revised manuscript will be part of a special issue comprising several manuscripts with a focus on different aspects of the described experiment. Since most of the experiments are based on a division of the experiment in phases as described by Paul et al. (2015), we decided to give a short description of these phases to avoid confusions with all other manuscripts. This phase division by Paul et al. (2015) was solely based on Chl a and temperature, which does not always match

bacterial parameters or changes in particulate and dissolved nutrient pools. Therefore, we intended to use a different phase division based on major changes in bacterial biovolume. However, we understand that a general division in temporal phases might be difficult. Hence, we decided to avoid a phase division for all statistical analyses. This paragraph will be reworked to focus clearly on bacteria and the heterotrophic processes of the experiment.

REVIEWER COMMENT 7: Lines 228-229 "During P2, concentrations of Chl a increased again". I don't think this concurs with the graph.

Author's response: We agree that the text was not clear and thus it will be improved in the revised manuscript.

REVIEWER COMMENT 8: Lines 236-237 A Spearman rank correlation does not allow to make an interpretation that distinguishes some treatments from others.

Author's response: We agree on that. The description in lines 236-237 is only based on a graphical evaluation. The text will be revised accordingly.

REVIEWER COMMENT 9: Lines 238-240 This negative relationship between BV of picos and Chl a is puzzling, especially since BV makes out the majority of phytoplankton biomass during the second half of the experiment.

Author's response: The relationship between BV of picophytoplankton and total Chl a does not reflect the total amount of Chl a or the contribution of picophytoplankton on total Chl a. At t13-t17 picophytoplankton contributed to ca. 50% of the total Chl a, but it's contribution increased from t17-t22 up to ca. 80% and stayed between ca. 80-100% upon the end of the experiment. In parallel, Chl a decreased after t17-t22 and stayed low until the end of the experiment. Therefore, BV of picophytoplankton and Chl a are negatively correlated during this period. However, we realized that we have to clarify this relationship more detailed and will amend the test and statistics accordingly.

REVIEWER COMMENT 10: Since bacteria are the focus of this manuscript (as I understand the introduction), the results regarding bacteria should be placed first, not phytoplankton.

Author's response: Since heterotrophic processes, mediated by bacteria are dependent on nutrient conditions as well as autotrophic processes mediated by phytoplankton, we intended to describe nutrients and phytoplankton first. However, we realize that changing this order would help to better focus the manuscript on bacteria. We will revise the manuscript accordingly.

REVIEWER COMMENT 11: The effects of the treatments on the bacterial variables throughout the experiment are very small. The only statistical effects reported are for P1 and by looking at the graphs (Fig. 3), the relationships with CO2 are hard to discern. Then a few time points are selected and emphasized in the results and discussion because they show differences in relation to CO2 treatments, but they make out a short period of the experiment.

Author's response: Although effects of the treatment on bacterial variables are small and only present for short time periods, they might have a huge impact on oceanic carbon cycling. Largest differences between the CO2-treatments on bacterial protein production (BPP) were measured after the breakdown of the Chl a maximum at t17, when BPP reached highest values throughout the experiment. During such periods, which are usually short in time, a relatively high turnover of organic matter occurs in natural systems. Therefore, these periods are of large importance for remineralisation processes and the carbon export. We thanks reviewer 1 for highlighting this issue and will give a better reasoning for choosing such periods in time. Especially, when direct effects of CO2 on bacterial variables are not expected, direct effects of CO2 on phytoplankton and nutrient pools might then indirectly feedback on bacterial variables during such periods of high organic matter turnover, when bacteria are most likely favoured and the bacterial metabolism is stimulated. We realized that this might have been not stated clear enough and will amend the revised version of the manuscript accordingly.

REVIEWER COMMENT 12: Figure 4 is not commented on in the results text?

Author's response: We will relate data presented in figure 4 to phytoplankton and bacterial biomass in the results section.

REVIEWER COMMENT 13: The discussion overall is a little tough to follow, since is not very closely aligned to or focused on the main issue. The discussion shows the difficulties in knowing what a statistical relationship means in this kind of study - the relative role of resource abundance, grazing and viral infections can only be speculated around. Still there are plenty of statements like "...revealed several indirect responses to fCO2, resulting from alterations in phytoplankton community composition and biomass". I am not convinced that the data support such statements.

Author's response: Unfortunately, we did not perform additionally experiments to justify the role of resource limitation (C/N/P), mixotrophy, or viral infections after day 25. We recognise that statements on those topics, which are not supported by measurements will certainly remain speculative. Therefore, we will focus in the revised version of the manuscript on aspects, which are well supported by data and try to remove any speculative statements.

References

Brussaard, C. P. D.: Optimization of procedures for counting viruses by flow cytometry, Appl. Environ. Microb., 70, 1506–1513, doi:10.1128/AEM.70.3.1506-1513.2004, 2004.

Crawfurd, K. J., Brussaard, C. P. D., and Riebesell, U.: Shifts in the microbial community in the Baltic Sea with increasing CO2, Biogeosciences Discuss., doi:10.5194/bg-2015-606, in review, 2016.

Legendre, P. and Anderson, M.J.: Distance-based redundancy analysis: testing multispecies responses in multifactorial ecological experiments. Ecological Monographs, 69, 1-24, 1999.

Mojica, K. D. A., Evans, C., and Brussaard, C. P. D.: Flow cytometric enumeration
of marine viral populations at low abundances, Aquat. Microb. Ecol., 71, 203–209, doi:10.3354/ame01672, 2014.

Paul, A., Bach, L.T., Schulz, K.-G., Boxhammer, T., Czerny, J., Achterberg, E.P., Hellemann, D., Trense, Y., Nausch, M., Sswat, M., and Riebesell, U.: Effect of elevated $CO_2$ on organic matter pools and fluxes in a summer Baltic Sea plankton community, Biogeosciences, 12, 6181–6203, doi:10.5194/bg-12-6181-2015, 2015.

---

## Author Response (AR2)

Dear Editor,

We are grateful for the numerous constructive comments on our manuscript from the two referees. Please find below our point by point responses to each referee comment and suggestion, as well as a revised version of our manuscript with and without track changes.

We revised our statistical analyses specifically by applying different multivariate approaches (e.g, Permutational multivariate analysis of variance (PERMANOVA); Distance-based linear modeling (DistLM); Distance-based redundancy analysis (dbRDA); principal component analysis (PCA); cluster analyses). Thereby we were able to account for potential interactions of several variables as proposed by the reviewers. All analyses were performed on entire data sets of physicochemical, metabolic or community variables. Phase-divisions were removed and the method section reworked according the reviewer´s suggestions. Thus, large parts of the manuscript were substantially revised. Thereby, we focus more specifically on bacterial variables and the coupling of bacteria to phytoplankton.

We are currently formatting the data files to be uploaded to the PANGAEA data base.

We thank you for the opportunity to submit a revised manuscript for consideration in Biogeosciences and look forward to hearing a response on the manuscript soon.

Yours Sincerely,

Thomas Hornick, on behalf of all authors

**Response to Reviewer #1**

We thank reviewer #1 for the constructive comments on our manuscript. Our responses to reviewer comments, including modifications to the manuscript, are detailed in the following:
* * *
**REVIEWER COMMENT 1:** This manuscript addresses an interesting, relevant and timely issue - how bacteria and their C processing may be affected by ocean acidification. As is also pointed out, there are no reasons to expect strong direct effects, while there may be indirect effects channelled through other parts of the food web. This topic is addressed in large scale mesocosms with differing levels of $CO_2$. Unfortunately, I don't find that the manuscript is very clear or efficient in addressing the issue. It is a difficult approach to study a large suite of variables that are to a large extent interdependent and try to understand what has actually happened. In my view, this study shows very minor (if any) effects of $CO_2$ on the bacterial variables measured, and it is hard to clearly link those minor effects to any particular process. Linguistically, I think the manuscript is clear, but I think results are overstated and relationships over-interpreted, and that the paper lacks a clear focus and structure.

**Author`s response:** We acknowledge that reviewer 1 raised these critical points. In contrast to most other studies dealing with effects of ocean acidification, we did not add nutrients to study the effects of changing $CO_2$ on nutrient cycling in a plankton community at naturally low nutrient conditions. The purpose of the experiment was to especially test effects of changes in $CO_2$ on a nutrient limited phytoplankton community and if possible effects on this phytoplankton community can feed back on bacterial activity and abundance. No pronounced direct effects of $CO_2$ on bacterial variables were observed throughout the experiment. Although only minor effects could be observed in this study, the obtained results will be crucial to better understand the role of nutrients on both direct and indirect effects of $CO_2$ on planktonic communities. However, we realized that some reported effects might be overemphasized in our old discussion and thus reconsidered their relevance. In the revised version of the manuscript we focus better on bacterial aspects and try to link them more specifically to particular processes, supported by very thoroughly reanalysed statistics (see also comments by and our reply to reviewer #2). Consequently, large parts of the manuscript have been revised according to the suggestions of both reviewers. Further detailed descriptions on changes, which were amended to the manuscript, will be answered in the following responses on the comments raised by the reviewers.

**REVIEWER COMMENT 2:** It is unclear in the title what "trophic interaction" refers to

**Author`s response:** We realize that the title was not clear in that respect. Based on our reanalyzed statistics and addressing specifically the coupling of bacteria to phytoplankton, the title has been changed: "Ocean acidification impacts bacteria-phytoplankton coupling at low nutrient-conditions."

**REVIEWER COMMENT 3:** There is too little information given to be able to evaluate the methods applied by reading this paper alone. There is a lot of self-referencing to papers covering the same experiment in all parts of the manuscript and this is problematic. Important information that is missing in the methods is for example the dimensions of the mesocosms and the principles behind measuring physical and chemical parameters.

**Author`s response:** Thanks for highlighting this important issue. In the old version, we reduced on purpose as much information as possible, which is given in the core paper by Paul et al. (2015) (i.e. measurements of dissolved and particulate nutrients) to condense our methods section and increase the word flow. However, we realized that it might be important to include brief descriptions on the measurement of physical and chemical parameters (Lines 134-175), metabolic parameters (Lines 233-245) as well as the mesocosm set-up (Lines 97-111) for providing a better background on the experiment, although this was already done in the core paper by Paul et al. (2015). In the revised manuscript we better described the methods and tried to reduce self-referencing to papers covering the same experiment wherever possible.

**REVIEWER COMMENT 4:** No information is given on the methods behind the estimation of low and high DNA bacteria. Results are included in the figures on low vs. high DNA bacteria, but not mentioned in the results text.

**Author`s response:** Two groups of heterotrophic prokaryotes were identified based on their low (LDNA) and high (HDNA) fluorescence. This identification was based on gating of SYBR green I fluorescence (nucleic-acid specific dye) against the side scatter signal determined by flow cytometry (Brussaard, 2004 with adaptation according to Mojica et al., 2014) as discussed in Crawfurd et al. (2015). We specified this in the revised manuscript (Lines 192-194) and mentioned observations in the ratio between LDNA and HDNA prokaryotes in the results section (Lines 302-305).

**REVIEWER COMMENT 5:** It is unclear how statistics were used to show the relationship between e.g. bacterial variables and $CO_2$ within a given time period - how did you account for time within each period?

**Author`s response:** So far, statistics were solely based on spearman rank correlation. Thereby, we assigned a spearman rank correlation between two variables using all measurements within a given time period. We realized (see rebuttal to reviewer #2), that this might be problematic for interpreting multivariate relationships. We revised the statistics specifically using multivariate approaches. Thereby we used permutational multivariate analysis of variance (PERMANOVA) to test for an effect of the $fCO_2$-treatment on chemical, metabolic and community data comprising entire datasets throughout the experiment. All phase-separations and applied statistics only comprising particular time-points were removed and data reanalyzed. Additionally we used distance-based redundancy

analysis (dbRDA) (Legendre and Anderson, 1999) for relating/modeling physicochemical variables (including temperature and PAR) to metabolic variable and microbial communities. To elucidate possible effects of the $f$CO$_2$-treatment on the co-occurence of different functional groups of the microbial communities, we performed cluster analyses on multiple spearman´s rank correlation coefficients. Thereby $p$-values were corrected for multiple comparisons. By applying multivariate approaches, we accounted for the temporal effect (i.e. two-factoral PERMANOVA with factors time and $f$CO$_2$-treatment).

(see section 2.5. Statistical analysis in the revised manuscript)

**REVIEWER COMMENT 6:** There is referencing in the results part. Lines 211-218 should be deleted. This manuscript should be able to stand on its own and not make the assumption that we have or will read the other papers from the same experiment. The motivation for dividing into P1 - P3 should be more explicit.

**Author`s response:** The revised manuscript will be part of a special issue comprising several manuscripts with a focus on different aspects of the described experiment. Since most of the experiments are based on a division of the experiment in phases as described by Paul et al. (2015), we decided to give a short description of these phases to avoid confusions with all other manuscripts. This phase division by Paul et al. (2015) was solely based on Chl $a$ and temperature, which does not always match bacterial parameters or changes in particulate and dissolved nutrient pools. Therefore, we intended to use a different phase division based on major changes in bacterial biovolume. However, we understood that a general division in temporal phases is difficult. Hence, we reanalyzed out statistics with multivariate approaches. All phase divisions were removed. We reworked the manuscript to focus clearly on bacteria and the trophic coupling of bacteria to phytoplankton at low nutrient conditions.

**REVIEWER COMMENT 7:** Lines 228-229 ”During P2, concentrations of Chl $a$ increased again”. I don't think this concurs with the graph.

**Author`s response:** The whole results section was substantially reworked, based on reanalyzed statistics. Most rather descriptive aspects were removed.

**REVIEWER COMMENT 8:** Lines 236-237 A Spearman rank correlation does not allow to make an interpretation that distinguishes some treatments from others.

**Author`s response:** We agree on that. The description in lines 236-237 is only based on a graphical evaluation. The whole statistical analyses have been revised (see Reviewer COMMENT 5).

**REVIEWER COMMENT 9:** Lines 238-240 This negative relationship between BV of picos and Chl *a* is puzzling, especially since BV makes out the majority of phytoplankton biomass during the second half of the experiment.

**Author`s response:** The relationship between BV of picophytoplankton and total Chl *a* does not reflect the total amount of Chl *a* or the contribution of picophytoplankton on total Chl *a*. At t13-t17 picophytoplankton contributed to ca. 50% of the total Chl *a,* but it´s contribution increased from t17-t22 up to ca. 80% and stayed between ca. 80-100% upon the end of the experiment (Paul et al., 2015). In parallel, Chl *a* decreased after t17-t22 and stayed low until the end of the experiment. Therefore, BV of picophytoplankton and Chl *a* are negatively correlated during this period. However, we realized that we had to clarify this relationship more detailed and addressed this relationship of picophytoplankton and bacterial biovolumes in section 4.1 (Lines 407-427).

**REVIEWER COMMENT 10:** Since bacteria are the focus of this manuscript (as I understand the introduction), the results regarding bacteria should be placed first, not phytoplankton.

**Author`s response:** Since heterotrophic processes, mediated by bacteria are dependent on nutrient conditions as well as autotrophic processes mediated by phytoplankton, we intended to describe nutrients and phytoplankton first. However, we realized that changing this order would help to better focus the manuscript on bacteria. We revised the results section accordingly, first describing statistical result and observations in the univariate data sets of bacterial variables and afterwards phytoplankton variables and then focusing on multivariate statistical approaches.

**REVIEWER COMMENT 11:** The effects of the treatments on the bacterial variables throughout the experiment are very small. The only statistical effects reported are for P1 and by looking at the graphs (Fig. 3), the relationships with $CO_2$ are hard to discern. Then a few time points are selected and emphasized in the results and discussion because they show differences in relation to $CO_2$ treatments, but they make out a short period of the experiment.

**Author`s response:** Although effects of the treatment on bacterial variables are small and only present for short time periods, they might have a huge impact on oceanic carbon cycling. Largest differences between the $CO_2$-treatments on bacterial protein production (BPP) were measured after the breakdown of the Chl *a* maximum at t17, when BPP reached highest values throughout the experiment. During such periods, which are usually short in time, a relatively high turnover of organic matter occurs in natural systems. Therefore, these periods are of large importance for remineralisation processes and the carbon export. Especially, when direct effects of $CO_2$ on bacterial variables are not expected, direct effects of $CO_2$ on phytoplankton and nutrient pools might then indirectly feedback on bacterial variables during such periods of high organic matter turnover, when bacteria are most likely favoured and the bacterial metabolism is stimulated. However, since Paul et al. (2016) did not report on changes

in carbon export across the study we reconsidered the importance of such observations during this particular study. We reanalyzed statistics, while focusing on consistent effects of $CO_2$ and the co-occurrence of functional groups of the microbial community.

**REVIEWER COMMENT 12:** Figure 4 is not commented on in the results text?

**Author`s response:** In the revised manuscript we report on all figures also in the results section. Further we improved the quality off all figures.

**REVIEWER COMMENT 13:** The discussion overall is a little tough to follow, since is not very closely aligned to or focused on the main issue. The discussion shows the difficulties in knowing what a statistical relationship means in this kind of study - the relative role of resource abundance, grazing and viral infections can only be speculated around. Still there are plenty of statements like "...revealed several indirect responses to $f$CO$_2$, resulting from alterations in phytoplankton community composition and biomass". I am not convinced that the data support such statements.

**Author`s response:** Unfortunately, we did not perform additionally experiments to justify the role of resource limitation (C/N/P), mixotrophy, or viral infections after day 25. We recognised that statements on those topics, which are not supported by measurements will certainly remain speculative. However, distance-based linear modeling (DistLM) and distance-based redundancy analysis (dbRDA) allowed to covering aspects like grazing, etc., by unexplained variance. Most speculative assumptions have been removed and the discussion has been substantially revised.

**REVIEWER COMMENT 4**: Throughout the manuscript, there are references to significant differences in values. However, there was only 1 mesocosm per $f$CO$_2$ level (except for duplicate controls), and no replicate sampling per mesocosm at each time point. There is no information about variation, and therefore, no statistical basis for making statements about significance. Declared differences are based on subjective assessments, rather than objective data analysis.

**Author´s response:** The reviewer raised an important point about the statistical analyses of the experiment. However, the experiment was designed to catch a gradient of different levels of CO$_2$ to apply regression analysis or having the opportunity to analyse tipping points of a response to CO$_2$ as well as analysing non-linear responses. We agree that we do not know a within-group variation of a single CO$_2$-treatment but this is not mandatory for regression analyses. Statistically, a regression is equally valid compared, i.e. to an analysis of variance (ANOVA) to making statements about significance. Besides, parameters with possible large measurement-variations or small sample volumes (i.e. bacterial protein production (BPP)) were measured in triplicate to account for the variance within the measurement. For these parameters the mean of 3 measurements is presented (i.e. see section 2.4). However, since these are pseudo-replicates, there is no additional value for any statistical test. We are aware that a spearman rank correlation is based on the rank and only describes the relationship between two variables by using a monotonic function. Therefore, it is probably not appropriate to make conclusions on multivariate interdependent variables. However, we reanalyzed the data and

applied more appropriate statistical approaches and models like dbRDA (see COMMENT 5 by reviewer #1).

**REVIEWER COMMENT 5:** The discussion could be more succinct and relevant. Much of section 4.2 can be removed, because it is mostly speculative, and ironically, emphasizes the confounding problem mentioned above. This section also contends that grazing was responsible for the drop in bacterial biovolume at higher $fCO_2$, but there is no supporting evidence from this study to support a grazing claim. This is an important point, because the claim is repeated in both the conclusion and abstract.

**Author´s response:** As reviewer 2 addresses right, final supporting data for any evidence of a grazing claim is missing. Therefore, we removed speculative assumptions and incorporated the section 4.2 into other sections of the discussion. The discussion has been reworked substantially.

**REVIEWER COMMENT 6:** Related to the decline in bacterial biovolume at higher $fCO_2$ are the actual results, displayed in Figure 2.I.C. Careful examination of that panel in the figure shows that one of the control mesocosms (368) exhibited a similar decline, for a slightly shorter period of time. In reality, without any information on variation around the data points, it is dangerous to be developing and discussing elaborate explanations of these patterns, if they are even accurate patterns.

**Author´s response:** We thank the reviewer for pointing out that this was not examined sufficiently previously in the manuscript. As pointed out before, we reworked our statistical analyses and removed such solely graphical interpretations.

**REVIEWER COMMENT 7:**

Minor points: Discussion: Numbering for the sections need to be corrected. There is no number for the first portion, and two sections labeled "4.1".

Figure 3. y-axis label for Figure 2.I.B should be for cell-specific BPP.

**Author`s response:** These 2 points have been corrected accordingly.

26  Maulbeerallee 2, Germany}

27  Correspondence to: T. Hornick (hornick@igb-berlin.de)

**Abstract**

 The oceans absorb about a quarter of the yearly produced anthropogenic atmospheric carbon dioxide ($CO_2$), resulting in a decrease in surface water pH, a process termed ocean acidification (OA). Surprisingly little is known about how OA affects the physiology  of heterotrophic bacteria or the coupling of heterotrophic bacteria to phytoplankton when  nutrients are limited. Previous experiments were  for the most part, undertaken during productive phases or following nutrient additions designed to stimulate algal blooms. Therefore, we undertook an *in situ* large-volume mesocosm (~55 m$^3$) experiment in the Baltic Sea by simulating different fugacities of $CO_2$ ($fCO_2$) extending from present to future conditions. The study was conducted in July-August after the nominal spring-bloom in  order to maintain low-nutrient conditions throughout the experiment. This resulted in  phytoplankton communities dominated by small-sized functional groups (picophytoplankton)). There was no consistent $fCO_2$-induced effect on Bacterial Protein Production (BPP), cell-specific BPP (csBPP) of ~~FL heterotrophic bacteria could not be explained exclusively by the availability of phytoplankton-derived organic carbon. The dynamics were also related to enhanced grazing on DNA rich (HDNA) bacterial cells at higher $fCO_2$ as revealed by flow cytometry. Additionally, a decoupling of autotrophic production and heterotrophic consumption during the last third of the experiment resulted in low, but significantly higher accumulation of DOC at enhanced $fCO_2$. Interestingly we could not detect any consistent and direct $fCO_2$-induced effect on BPP, csBPP nor BV. In contrast, our results reveal several indirect $fCO_2$-
[revised manuscript text omitted]

| | | FL size | | | | PA size | | |
|---|---|---|---|---|---|---|---|---|
| | | BV$_{HP}$ | BPP | esBPP | | BV$_{HP}$ | BPP | esBPP |
| $f$CO$_2$ | | P0: - | P0: - | P0: - | | P0: - | P0: - | P0: - |
| DOC | | P0: - | P0: - | P0: - | | P0: - | P0: - | P0: - |
| CR | | P0: -0.71; <0.01; 12
P1: 0.58; <<0.01; 42
P2: 0.64; <<0.01; 106
P3: 0.59; <<0.01; 36 | P0: n.s.
P1: n.s.
P2: 0.72; <<0.01; 36
P3: n.s. | P0: n.s.
P1: n.s.
P2: 0.51; <0.01; 36
P3: n.s. | | P0: -0.62; 0.03; 12
P1: 0.5; 0.03; 18
P2: 0.5; <0.01; 36
P3: n.s. | P0: n.s.
P1: n.s.
P2: 0.71, <<0.01; 36
P3: n.s. | P0: n.s.
P1: n.s.
P2: n.s.
P3: n.s. |
| Chl $a$ | | P0: n.s.
P1: 0.77; <<0.001; 48
P2: -0.77;<<0.001; 112
P3: n.s. | P0: -0.59; 0.04; 12
P1: 0.48; 0.02; 24
P2: -0.41; <0.01; 41
P3: n.s. | P0: -0.89; 0.02; 6
P1: n.s.
P2: n.s.
P3: n.s. | | P0: -0.65; 0.02; 12
P1: 0.39; 0.05; 24
P2: n.s.
P3: n.s. | P0: n.s.
P1: 0.51; 0.01; 24
P2: -0.49, <0.01; 41
P3: n.s. | P0: n.s.
P1: n.s.
P2: -0.41; 0.01; 41
P3: -0.31; 0.05; 41 |
| BV$_{Nano}$ | | P0: n.s.
P1: n.s.
P2: -0.75; <<0.01; 112
P3: -0.46; <<0.01; 51 | P0: n.s.
P1: n.s.
P2: -0.35; 0.02; 42
n.s. | P0: n.s.
P1: n.s.
P2: n.s.
P3: 0.35; 0.05; 33 | | P0: n.s.
P1: n.s.
P2: n.s.
P3: -0.32; 0.05; 39 | P0: 0.83; 0.04; 6
P1: n.s.
P2: -0.44, <<0.01; 42
P3: n.s. | P0: n.s.
P1: n.s.
P2: 0.34; 0.03; 42
P3: n.s. |
| BV$_{Pico}$ | | P0: 0.74; <0.01; 12
P1: 0.79; <<0.01; 48
P2: 0.91; <<0.01; 112 | P0: n.s.
P1: 0.52; <0.01; 24 | P0: n.s.
P1: n.s.
P2: n.s. | | P0: n.s.
P1: 0.71; <<0.01; 24 | P0: n.s.
P1: 0.58; <0.01; 24
P2: 0.73, <<0.01; 42 | P0: n.s.
P1: n.s.
P2: |

| | | | | | |
|---|---|---|---|---|---|
| | P3: n.s. | P2: 0.65; <<0.01; 42
P3: n.s. | P3: n.s. | P2: 0.31; 0.04; 42
P3: n.s. | P3: n.s. | 0.37; 0.01; 42
P3: n.s. |
| $BV_{Syn}$ | P0: 0.87; <<0.01; 12
P1: 0.86; <<0.01; 48
P2: 0.89; <<0.01; 112
P3: n.s. | P0: n.s.
P1: 0.5; 0.01; 24
P2: 0.56; <<0.01; 42
P3: -0.44; <0.01; 38 | P0: n.s.
P1: n.s.
P2: n.s.
P3: -0.47; <0.01; 33 | P0: n.s.
P1: 0.64; <<0.01; 24
P2: n.s.
P3: n.s. | P0: 0.83; 0.04; 6
P1: 0.55; <<0.01; 24
P2: 0.55, <<0.01; 42
P3: -0.5; <0.01; 38 | P0: n.s.
P1: n.s.
P2: 0.37; 0.01; 42
P3: n.s. |
| $BV_{PicoI}$ | P0: 0.9; <<0.01; 12
P1: 0.82; <<0.01; 48
P2: 0.36; <<0.01; 110
P3: -0.28; 0.05; 51 | P0: n.s.
P1: 0.64; <<0.01; 24
P2: n.s.;
P3: n.s. | P0: n.s.
P1: 0.53; <0.01; 24
P2: n.s.
P3: -0.34; 0.05; 33 | P0: n.s.
P1: 0.6; <0.01; 24
P2: n.s.
P3: n.s. | P0: n.s.
P1: 0.65; <<0.01; 24
P2: n.s.
P3: n.s. | P0: 0.83; 0.04; 6
P1: n.s.
P2: n.s.
P3: n.s. |
| $BV_{PicoII}$ | P0: -0.76; <0.01; 12
P1: 0.6; <<0.01; 48
P2: n.s.;
P3: 0.36; 0.01; 51 | P0: n.s.
P1: 0.54; <0.01; 24
P2: n.s.
P3: 0.46; <0.01; 38 | P0: n.s.
P1: 0.4; 0.05; 24
P2: n.s.
P3: | P0: n.s.
P1: 0.58; <0.01; 24
P2: 0.54; <<0.01; 42
P3: n.s. | P0: 1; <<0.01; 6
P1: 0.63; <0.01; 24
P2: n.s.
P3: n.s. | P0: 0.94; <0.01; 6
P1: n.s.
P2: n.s.
P3: n.s. |
| $BV_{PicoIII}$ Res | P0: n.s. | P0: n.s. | P0: n.s. | | P0: n.s. | P0: n.s. | P0: n.s. |
| $BV_{NanoI}$ Total | P0: n.s. | P0: n.s. | P0: n.s. | | P0: n.s. | P0: 1; | P0: |
| $BV_{NanoII}$ | P0: n.s.
P1: n.s.
P2: -0.76; <<0.01; 112
P3: n.s. | P0: n.s.
P1: n.s.
P2: -0.37; 0.02; 42
P3: n.s. | P0: n.s.
P1: n.s.
P2: n.s.
P3: n.s. | P0: n.s.
P1: n.s.
P2: n.s.
P3: n.s. | P0: 0.81; 0.05; 6
P1: n.s.
P2: -0.46; <<0.01; 42
P3: n.s. | P0: n.s.
P1: n.s.
P2: -0.34; 0.03; 42
P3: n.s. |

[revised manuscript text omitted]

[*] variables selected in step-wise procedure based on AIC.